# A sporulation signature protease is required for assembly of the spore surface layers, germination and host colonization in *Clostridioides difficile*

Eleonora Marini[1], Carmen Olivença[1], Sara Ramalhete[1], Andrea Martinez Aguirre[2], Patrick Ingle[3], Manuel N. Melo[1], Wilson Antunes[1], Nigel P. Minton[3], Guillem Hernandez[1], Tiago N. Cordeiro[1], Joseph A. Sorg[2], Mónica Serrano[1], Adriano O. Henriques[1] *

1 Instituto de Tecnologia Química e Biológica António Xavier, Universidade Nova de Lisboa, Avenida da República EAN, Oeiras, Portugal, 2 Texas A&M University, Department of Biology, College Station, Texas, United States of America, 3 Clostridia Research Group, BBSRC/EPSRC Synthetic Biology Research Centre (SBRC), School of Life Sciences, University of Nottingham, Nottingham, United Kingdom

* aoh@itqb.unl.pt

**Data Availability Statement:** All relevant data are within the manuscript and its Supporting Information files.

## Abstract

A genomic signature for endosporulation includes a gene coding for a protease, YabG, which in the model organism *Bacillus subtilis* is involved in assembly of the spore coat. We show that in the human pathogen *Clostridioidesm difficile*, YabG is critical for the assembly of the coat and exosporium layers of spores. YabG is produced during sporulation under the control of the mother cell-specific regulators $\sigma^E$ and $\sigma^K$ and associates with the spore surface layers. YabG shows an N-terminal SH3-like domain and a C-terminal domain that resembles single domain response regulators, such as CheY, yet is atypical in that the conserved phosphoryl-acceptor residue is absent. Instead, the CheY-like domain carries residues required for activity, including Cys207 and His161, the homologues of which form a catalytic diad in the *B. subtilis* protein, and also Asp162. The substitution of any of these residues by Ala, eliminates an auto-proteolytic activity as well as interdomain processing of CspBA, a reaction that releases the CspB protease, required for proper spore germination. An in-frame deletion of *yabG* or an allele coding for an inactive protein, *yabG^{C207A}*, both cause misassemby of the coat and exosporium and the formation of spores that are more permeable to lysozyme and impaired in germination and host colonization. Furthermore, we show that YabG is required for the expression of at least two $\sigma^K$-dependent genes, *cotA*, coding for a coat protein, and *cdeM*, coding for a key determinant of exosporium assembly. Thus, YabG also impinges upon the genetic program of the mother cell possibly by eliminating a transcriptional repressor. Although this activity has not been described for the *B. subtilis* protein and most of the YabG substrates vary among sporeformers, the general role of the protease in the assembly of the spore surface is likely to be conserved across evolutionary distance.

**Funding:** This work was supported by the European Union Marie Sklodowska Curie Innovative Training Networks (contract number 642068) to AOH and EM was the recipient of a PhD fellowship under that contract. This project was supported by awards PTDC/BIA-MIC/29293/2017 to MS from FCT ("Fundação para a Ciência e a Tecnologia") and 5R01AI116895 and 1U01AI124290 to J.A.S. from the National Institute of Allergy and Infectious Diseases. The content is solely the responsibility of the authors and does not necessarily represent the official views of the NIAID. The funders had no role in study design, data collection and interpretation, or the decision to submit the work for publication. A.M.A. was supported by the Mexican Science and Technology Council (CONACYT Mexico) under award number 625561/472087. This work was also financially supported by Project LISBOA-01-0145-FEDER-007660 ("Microbiologia Molecular, Estrutural e Celular") funded by FEDER funds through COMPETE2020 – "Programa Operacional Competitividade e Internacionalização" (POCI), by national funds through the FCT. The content is solely the responsibility of the authors and does not necessarily represent the official views of the NIAID. The funders had no role in study design, data collection and analysis, decision to publish, or preparation of the manuscript.

**Competing interests:** The authors have declared that no competing interests exist.

## Author summary

*Clostridioides difficile*, an anaerobic spore-forming bacterium, colonizes the gastro-intestinal tract when, as during antibiotic treatment, the protective effect of the microbiota is disrupted. A leading agent of nosocomial infections, causing a range of symptoms from mild diarrhea to life-threatening conditions, the organism is recognized as a global and urgent threat. Infection begins with the ingestion of spores, which will germinate in response to bile salts. Two proteinaceous spore surface layers, the coat and the exosporium, play a crucial role in infection and colonization, as they contribute to spore resistance, binding to host cells and the interaction with and the response to germinants. The *yabG* gene, part of a genomic signature for sporulation, codes for a cysteine protease, with residues required for catalysis embedded in a CheY-like response regulator receiver domain. YabG is required for proper morphogenesis of the spore surface layers, germination and host colonization. YabG also regulates the mother cell line of gene expression by allowing the expression of genes required for assembly of the coat and exosporium. While this latter function has not been described for other organisms, the general role of *yabG* in the assembly of the spore surface layers is likely to be conserved among spore-formers.

## Introduction

Able to colonize the gastro-intestinal tract when the protective effect of the microbiota is disrupted, *Clostridioides difiicile* [1] is the leading cause of nosocomial diarrhoea linked to antibiotic therapy. Infection can, however, lead to more serious complications, including pseudomembranous colitis, toxic megacolon, bowel perforation and in the most severe cases sepsis and death [2–4]. Changes in the epidemiology of *C. difficile* are causing increased incidence in the community and the risk of zoonotic transmission is an additional threat [2,4–8]. Two large toxins, TcdA and TcdB, are the main virulence factors and the direct cause of the disease symptoms [9,10].

A strict anaerobe, *C. difficile* relies on spore formation for dissemination and environmental persistence [2–4]. Infection starts with the ingestion of spores that will germinate in the small intestine in response to certain bile salts; at least a fraction of the vegetative cells that outgrow from spores will produce the TcdA and TcdB toxins, and some will differentiate into spores [9,11–14]. Spores have a central compartment harbouring the chromosome, delimited by a membrane and surrounded by a thin layer of peptidoglycan that becomes the wall of the cell resulting from spore germination. This unit is enclosed in a much thicker layer of modified peptidoglycan, called the spore cortex. The cortex, essential for spore dormancy, is in turn covered by several proteinaceous layers that together form the surface of spores; the structure and composition of these layers differs greatly among sporeformers [15,16]. In *B. subtilis* the spore surface consists of a glycosylated crust tightly adherent to an underlying multi-layered coat [17]. In the pathogens *B. cereus* and *B. anthracis*, the coat is surrounded by an exosporium separated from the coat by an interspace [15,16]. The coat/crust afford protection against cortex-lytic enzymes, such as lysozyme, and against small molecules such as oxidizing agents. The exosporium provides physical robustness and serves as a permeability barrier that excludes enzymes and antibodies. Both the coat/crust and the exosporium also affect the interaction of spores with germinants and mediate the interactions of spores with host cells and abiotic surfaces [16,18].

In *C. difficile*, an exosporium tightly adherent to the underlying coat, forms the spore outermost structure [19–22]. Proper assembly of the coat and exosporium is important for colonization and infection. Components of the coat and exosporium have been identified that

highlight the role of these structures in the interaction of spores with the colonic mucosa, colonization and virulence [23–25]. These include the cysteine-rich proteins CdeC and CdeM; *cdeC* and *cdeM* mutants, which have a misassembled coat and exosporium, show altered colonization and virulence [20,24–26]. Importantly, recent work has shown that the interaction of spores with E-cadherin promotes spore binding to and internalization by intestinal epithelial cells, which in turn contributes to infection recurrence [27,28]. Clearly, the identification and functional characterization of proteins that govern assembly of the coat and exosporium layers remains an important research goal that will inform us on the role and behaviour of spores during the initial stages of infection.

At the onset of sporulation, the rod-shaped cell divides asymmetrically to form a large mother cell and a smaller forespore. The genetic programs that are then activated in the two cells are governed by a cascade of cell type-specific RNA polymerase sigma factors, $\sigma^F$ and $\sigma^G$ in the forespore and $\sigma^E$ and $\sigma^K$ in the mother cell [29–31]. $\sigma^F$ and $\sigma^E$ control early stages of development, whereas $\sigma^G$ and $\sigma^K$ are active mainly when the mother cell completes engulfment of the forespore, to produce a cell within a cell. Formation of the spore surface structures is mainly a function of the mother cell and requires both $\sigma^E$ and $\sigma^K$ [32,33]. Using the known *B. subtilis* sporulation genes, a core machinery for sporulation was identified, and a genomic signature defined as those genes present in at least 95% of the genomes of organisms known to sporulate, and in less than 5% of other bacterial genomes [34]. Strikingly, other than four vegetative genes that are co-opted for sporulation and expressed from $\sigma^K$-dependent promoters, the $\sigma^K$-regulon contributed to the genomic signature with a single gene, *yabG* [34]. While this reflects the diversity of the genes coding for components of the spore surface layers among sporeformers, it also hinted at an important, phylogenetically conserved function for *yabG*. As its *B. subtilis* counterpart, the *yabG* gene of *C. difficile* is under the control of $\sigma^K$ [30,31,35].

In *B. subtilis* and in *B. anthracis*, a role for *yabG* in the assembly of the spore coat has been shown [36–41] and in *C. difficile*, *yabG* plays a part in spore germination ([42,43]; see also below). In *B. subtilis* YabG associates with the spore coat and is required for cleavage of at least six spore coat proteins [36–39]. Two of those proteins, SafA and C30, which have important morphogenetic functions in assembling of the inner coat layers, are cleaved in vitro by partially purified YabG [38]. The YabG-dependent cleavage of SafA, C30 and two other inner coat proteins, is important for their subsequent cross-linking by a coat-associated transglutaminase [38]. These substrates of *B. subtilis* YabG are not found in *C. difficile* [34,44]. Another likely substrate of *B. subtilis* YabG is SpoIVA, a conserved morphogenetic ATPase required for the formation of a basal layer upon which the coat/crust/exosporium are assembled [31,45], SpoIVA may also be a YabG substrate in *C. difficile* because the protein is found at significantly higher levels in coat extracts from spores of a *yabG* mutant [42]. Two other proteins, not found in *Bacillus* sporeformers, also accumulate in their unprocessed forms in spores of a *C. difficile yabG* mutant, CspBA and Pre-pro-SleC [42,43,44]. Interdomain processing of CspBA, releases CspB, whereas cleavage of Pre-pro-SleC produces pro-SleC [42,43]. CspB is a subtilisin-like serine protease involved in the activation, together with the germinant receptor CspC, of pro-SleC [46,47]. SleC, in turn, is a cortex hydrolase essential for spore germination in response to bile salts [47–51]. Processing of Pre-pro-SleC and interdomain processing of CspBA requires *yabG* [52]. YabG also resulted in processing of SleC^FL to pro-SleC in *E. coli* extracts, and a processing site was tentatively identified in CspBA which is also a direct substrate of the protease [42,43]. CspA is important for recognition of co-germinants and mutations in *yabG* were found to render germination in response to the bile salt taurocholate independent of co-germinants such as glycine [43].

We show that *C. difficile* YabG has a C-terminal domain that resembles a single domain response regulator such as CheY. Residues within the CheY-like domain are required for an

auto-proteolytic activity that leads to its complete degradation and for cleavage of its substrates. These residues include C207 and H161 which occupy positions homologous to those shown to form a catalytic diad required for auto-proteolysis in *B. subtilis* YabG [52]. We show that YabG is recruited to the developing spore and that its assembly is temporally controlled by auto-proteolysis. We show that YabG governs attachment of the coat to the underlying cortex peptidoglycan, formation of the exosporium and is also involved in germination in line with its recently demonstrated role in regulating the sensitivity of *C. difficile* spores to co-germinants [43]. Importantly, a *yabG* mutant is impaired in host colonization. Finally, we show that YabG is also required for the expression of a late class of σ$^K$-dependent genes involved in coat/exosporium assembly, thus contributing to the control of the mother cell line of gene expression. YabG defines a novel type of cysteine (thiol) protease dedicated to the assembly of the spore surface layers in sporeforming organisms.

## Results

### YabG is a cysteine protease with a CheY-like fold

In *C. difficile*, *yabG* (annotated as CD630_35690 in the genome of strain 630), is followed by a homologue of the *B. subtilis veg* gene, itself just upstream of *sipL*, coding for a protein essential for assembly of the coat/exosporium layers of spores [53,54] (Fig 1A). In *B. subtilis*, *veg*, which is also downstream of *yabG* (S1A Fig), was shown to have a role in biofilm formation and is expressed in both vegetative and sporulating cells [55,56]. Recent work showed that *B. subtilis* YabG is a cysteine protease and suggested that it uses a catalytic dyad formed by Cys218 and His172 in an auto-proteolytic activity that causes its rapid degradation (Fig 1B) [52]. *B. subtilis* YabG, together with the *C. difficile* protein and 202 other orthologues from sporeformers, was included in a new family of clan CD in the MEROPS peptidase database [52,57].

An alignment of the residues around Cys218 of the *B. subtilis* YabG protein with the corresponding region from selected orthologues, reveals a highly conserved region with an invariant Cys residue, C207, in *C. difficile* YabG (Fig 1B). Likewise, His161 in the *C. difficile* protein, homologous to His172 in *B. subtilis* YabG, is also invariant among YabG orthologues. Thus, the two catalytic residues suggested for *B. subtilis* YabG are conserved. We noticed, however, that an Asp residue, at position 162 in the *C. difficile* protein, is also strictly conserved (Fig 1B). Cysteine proteases share an acylation-deacylation mechanism involving a nucleophilic cysteine thiol that is part of a Cys-His dyad or a Cys-His-Asp/Asn catalytic triad [58,59]. Thus, Asp162 could also be important for activity.

To gain more insight into the structure and function of YabG, we generated and refined a structural model using the machine learning approach to protein structure prediction implemented through AlphaFold2 [60]. The model generated for *C. difficile* YabG suggests that the protein consists of two independent structural domains, A (Fig 1C, residues 1–57, in orange) and B (residues 99–286, in green) connected by a linker, L (residues 58–98, in blue) (see also S2 Fig). A search using the Fold Seek server [61] indicates structural similarity of domain A to proteins with an Src homology domain 3 (SH3), including for example the cyanobacterial protein PetP and *Escherichia coli* HspQ. PetP is a subunit of cytochrome *b6f* [62], while HspQ is both a substrate and a specificity and allosteric enhancing factor for the Lon protease [63,64] (S3 Fig). SH3 domains are small, 55–70 residues-long protein-protein interaction modules [65,66].

Fold Seek also revealed structural similarity of B to the receiver domain of response regulators including, among the top hitters, the KdpE protein of *E. coli* (pdb identifier: 4I85), MicA from *Streptococcus pneumoniae* (7m0s), MtrA from *Mycoacterium tuberculosis* (3nhz) and CheY from *Vibrio cholerae* (4hnr) and *Thermotoga maritima* (4tmy). All of these proteins

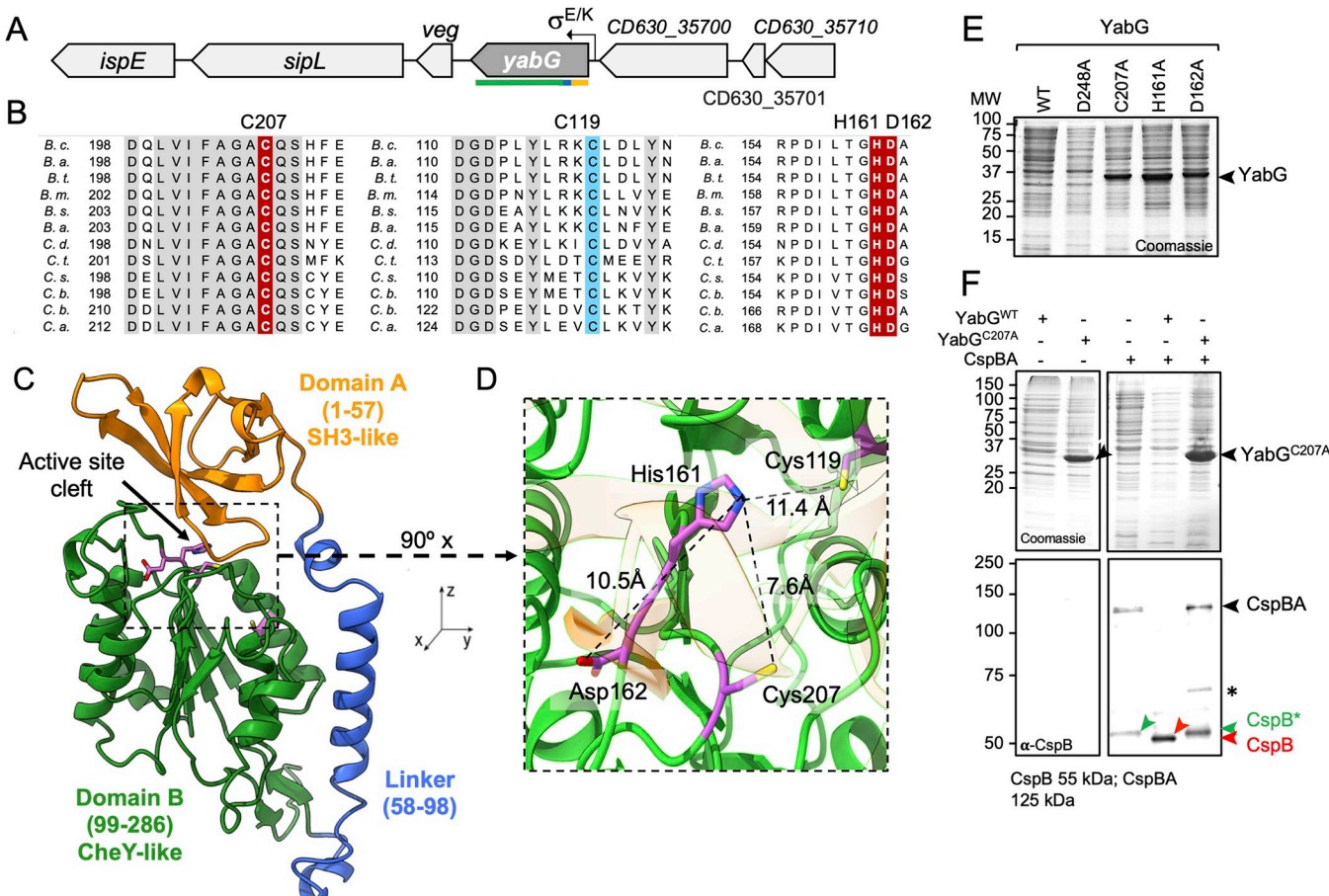

**Fig 1. The *yabG* region of the *C. difficile* chromosome and organization of YabG. A**: Schematic representation of the *yabG* region of the *C. difficile* chromosome. A σ^E/K^-dependent promoter in front of the *yabG* gene is represented. The lines below the gene represent the regions coding to distinct regions of the YabG protein (as in panel C). **B**: The alignments show highly conserved or invariant residues (grey) in the vicinity of Cys residues at positions 119 (blue) and 207 of YabG (brown) and around His161 and Asp162 (brown). **C**: An AlphaFold2 model of *C. difficile* YabG. The model highlights a N-terminal SH3-like domain (A, orange; residues 1–57) and a C-terminal CheY-like domain (B, green; residues 99–286) connected by a linker (L, blue, residues 58–98). A putative active site cleft is indicated at the interface between A and B. **D**: Expansion of the active site region to show the relative positions of Cys207, His161, Asp162 and Cys119 with estimated distances shown in Å. **E** and **F**: The WT and the indicated forms of YabG and CspBA were produced in *E. coli*. In F, the indicated forms of YabG were produced together with or in the absence of CspBA The proteins in whole cell extracts were resolved by SDS-PAGE and the gels stained with Coomassie or subject to immunoblotting (in **F**) with anti-CspB antibodies. In **E** and **F**, the black arrowheads show the position of YabG, YabG^C207A^ or CspBA. The green arrowhead in **F** shows the position of CspB released from CspBA, independently of YabG, while the red arrowhead points to CspB released from CspBA through the action of YabG. The asterisk refers to an unspecific band.

share the fold of the archetypal CheY response regulator from *E. coli* [67–69]. The superimposition of the model obtained for domain B of YabG with the crystal structures of KdpE and CheY highlights the similarity to the CheY fold (S4 Fig). KdpE, MicA and MtrA have a receiver domain and a C-terminal DNA binding domain and function as transcription factors [70–72]. CheY, in turn, is a single domain response regulator which upon phosphorylation of an Asp residue, binds directly to the flagellar motor [67]. Most response regulators function through phosphorylation of a conserved aspartate residue [67–69] but others function independently of phosphorylation [68,69]. The phospho-acceptor residue in CheY is D57; this residue is not conserved in YabG with the homologous position occupied by Thr159 in *C. difficile* YabG and by Thr164 in the *B. subtilis* protein (Fig 1B). Thus, YabG may function independently of phosphorylation.

## Residues of YabG involved in auto-proteolysis

The AlphaFold2 model suggests that the catalytic Cys207 in B is at the bottom of a cleft formed between the A domain and the top of B (Fig 1C). The distances estimated between Cys207 and His161 from the model, and between the latter and Asp162 (Fig 1D) are longer than in other Cys proteases and the side chain of His161 is not oriented towards Cys207 [58,59]. However, of the 204 YabG sequences in the MEROPS U57 family, where YabG used to be included [57], 96.1% have His, Asp and Cys residues conserved at equivalent positions. A second Cys residue at position 119 in the *C. difficile* protein is also invariant among YabG orthologues (Fig 1B).

To test whether *C. difficile* YabG showed auto-proteolytic activity and if so, what residues of the putative active site were involved, we overproduced YabG$^{WT}$ in *E. coli*, using an auto-induction regime [73], as a N-terminal His$_{10}$ fusion. His$_{10}$-YabG$^{WT}$ did not accumulate in whole cell extracts, as assessed by Coomassie staining (Fig 1E). In contrast, *B. subtilis* YabG accumulated but underwent auto-proteolytic degradation over time [52]. Why the *B. subtilis* appears more stable than its *C. difficile* counterpart is presently unknown. To test whether Cys207, His161 and Asp162 were required for YabG auto-proteolysis, YabG$^{C207A}$, YabG$^{H161A}$ and YabG$^{D162A}$ were also overproduced in *E. coli*. As assessed by Coomassie staining, the YabG$^{C207A}$, Yab-G$^{H161A}$ and YabG$^{D162A}$ forms of the protein accumulated in the extracts as species of about 34 kDa, consistent with the predicted size of the protein (34.7 kDa), indicating that the substitutions impaired auto-proteolysis (Fig 1E). In contrast, another single alanine substitution, D248A in domain B, did not cause the protein to accumulate in extracts (Fig 1E). Only substitutions that impair protease activity allow YabG to accumulate. Thus, YabG residues Cys207 and His161, homologous to the dyad described for the *B. subtilis* protein, and Asp162, are required for auto-proteolysis. We also tested whether a variant with Cys119 changed to Ala accumulated in extracts. In contrast to YabG$^{C207A}$, the YabG$^{C119A}$ form did not accumulate in whole cell extracts (S5A Fig). Thus, Cys119 is not required for the auto-proteolytic activity of YabG.

Auto-proteolysis of *B. subtilis* YabG leads to the accumulation of transiently stable fragments through cleavage after Arg5, Arg17, Arg49 and Arg93 [52]. Of these, Arg17 and Arg49 are conserved in *C. difficile* YabG (Arg9 and Arg41, respectively); the position homologous to Arg93 is occupied by a Lys in the *C. difficile* protein but there are two Arg´s in the vicinity of this residue (S5B Fig). This suggests that the *C. difficile* protein may also be cleaved at least after Arg9 and Arg41. That *C. difficile* YabG is likely to be specific for Arg at the P1 position is in line with cleavage of the SleC and CspBA after Arg residues [43].

## C207 is required for YabG-dependent CspBA interdomain processing

CspBA accumulates in spores of *yabG* mutants [42,43,74]. Release of the CspB domain from CspA is necessary for the activation of the cortex hydrolase SleC, required for germination completion [47,51]. YabG was found to be involved in CspBA processing [42,43]. To assess the role of Cys207 in the reaction, YabG$^{WT}$ or YabG$^{C207A}$ were co-produced in *E. coli* together with CspBA. Production of CspBA alone resulted in the accumulation of a species of about 125 kDa, consistent with the expected size of CspBA (MW 124.6 kDa), detected with anti-CspB antibodies (Fig 1F, black arrowhead in the bottom panel). A species of about 60 kDa was also detected (Fig 1F, green arrowhead). This species, termed CspB*, results from alternative processing that occured in a YabG-independent manner [42]. In this experiment, and as described above, only YabG$^{C207A}$ was detected (Fig 1F). However, even though YabG$^{WT}$ was not detected, its co-production with CspBA resulted in the disappearance of CspBA and the accumulation of a species of about 55 kDa, the size expected for the CspB moiety, detected with anti-CspB antibodies (Fig 1F, red arrowhead). Accumulation of this protein required the activity of YabG: the co-production of CspBA together with YabG$^{C207A}$ did not lead to

depletion of CspBA or to the accumulation of CspB, whereas the YabG-independent CspB*
still accumulated (Fig 1F, bottom panel).

We conclude that Cys207 is required for CspBA interdomain processing, consistent its pre-
dicted role in catalysis and with the accumulation of the precursor protein in *yabG* spores [42,43].

## *yabG* is expressed in the mother cell during late stages of spore morphogenesis

Synthesis of the proteins that form the spore coat and exosporium layers is driven by $\sigma^E$ and
$\sigma^K$, which are mainly active before and after engulfment completion, respectively [15,75]. In *B.
subtilis*, *yabG* is under the control of $\sigma^K$ [39] and genome-wide transcriptional profiling studies
of *C. difficile* sporulating cells have suggested that *yabG* is also under the control of $\sigma^K$ in this
organism [30,31,35]. Consistent with these studies, putative -10 and -35 promoter elements
that match the consensus for $\sigma^K$ recognition are present in the *yabG* regulatory region (S6A
Fig) [30,76]. To determine the time of *yabG* expression relative to the stages of spore morpho-
genesis, we made use of a transcriptional fusion between the *yabG* promoter region and the
$SNAP^{Cd}$ reporter [29,30,77]. The $P_{yabG}$-$SNAP^{Cd}$ fusion was introduced into the WT strain
630$\Delta erm$ as well as into congenic *sigE*::*erm* and *sigK*::*erm* mutants [29]. The resulting strains
were inoculated on 70:30 agar plates [53] and imaged by phase contrast and fluorescence
microscopy, following labelling with the SNAP substrate TMR-Star, 14 and 20 hours thereaf-
ter. In the WT, expression of $P_{yabG}$-$SNAP^{Cd}$ at hour 14 (S6B Fig, top panels) and at hour 20
(bottom panels) was confined to the mother cell. A fluorescence signal from $P_{yabG}$-$SNAP^{Cd}$
was undetected in the *sigE*::*erm* mutant, but still detected in the mother cell of a *sigK*::*erm*
mutant (S6B Fig). This suggests dual control of *yabG* expression by $\sigma^E$ and $\sigma^K$. The consensus
for $\sigma^E$ recognition is also included in S6A Fig; the highly conserved ATA motif for $\sigma^E$ recogni-
tion in the -35 region is absent, but the putative -10 region conforms better to the consensus
for $\sigma^E$ binding than to the consensus for $\sigma^K$ binding [76].

To determine the main period of *yabG* expression, we measured the expression of $P_{yabG}$-
$SNAP^{Cd}$ during spore morphogenesis. $P_{yabG}$-$SNAP^{Cd}$ expression was detected at 20 hours of
growth in 45% of the cells during asymmetric division and engulfment, in 46% of the sporan-
gia of phase-dark forespores, in 45% of the sporangia of phase-grey forespores, and in 53% of
sporangia of phase-bright forespores (S6B Fig, yellow arrowheads). The average fluorescence
intensity from $P_{yabG}$-$SNAP^{Cd}$ in sporangia of phase-bright forespores at hour 20 was higher
than that of sporangia of phase-dark or phase-grey spores or cells during asymmetric division
and engulfment (S6C Fig). At hour 14 the percentage of sporangia with a signal was highest for
sporangia of phase-dark spores and the intensity of the fluorescence signal per cell remained
relatively uniform regardless of the developmental stage (S6B and S6C Fig). While no expres-
sion was detected in a *sigE* mutant, the intensity of the fluorescence signal for $P_{yabG}$-$SNAP^{Cd}$ in
the *sigK*::*erm* mutant was decreased relative to the WT, particularly at hour 14 (S6C Fig). Nev-
ertheless, 90 percent of the sporangia of the *sigK* mutant at hour 14 and 74 percent at hour 20
showed a fluorescence signal presumably because of persistent activity of $\sigma^E$. Together, these
results indicate that the onset of *yabG* expression in the mother cell occurs soon after asym-
metric division, under the control of $\sigma^E$, when the mother cell starts engulfing the forespore; it
then increases towards the final stages of sporulation, with the contribution of $\sigma^K$, when the
forespore transitions from phase-dark/grey to phase-bright (S6 Fig)

## The activity of YabG is required for proper spore germination

Amino acids such as alanine or glycine act as co-germinants during spore germination trig-
gered by cholic acid derivatives as for instance taurocholate (TA) (reviewed in [78]). Previous

work has shown that deletion of or point mutations in *yabG* allowed spore germination in response to TA alone in epidemic strain R20291 [43]. To examine the role of YabG in the germination of *C. difficile* 630Δ*erm* spores, we constructed a congenic *yabG* in-frame deletion mutant, Δ*yabG*, using a CRISPR-Cas9 system (S7A and S7B Fig). We constructed two additional strains in the Δ*yabG* background, with either a copy of the WT *yabG* gene (*yabG*$^C$) or *yabG*$^{C207A}$ at the non-essential *pyrE* locus (S7C and S7D Fig). In the background of strain 630Δ*erm* we found that spores of the *yabG* mutants germinated slower than WT spores in response to TA in a rich medium (S8A, S8B and S9 Figs). Previous work has shown that the efficiency of germination for Δ*yabG* spores, as assessed by plating spores exposed to TA onto plates of a rich medium containing the germinant was 0.8 of the WT [42]. Using the same assay, we obtained a plating efficiency of 0.71±0.13 for Δ*yabG* spores, 0.89±0.13 for *yabG*$^{C207A}$ spores and 0.96±0.18 for *yabG*$^C$ spores (S8C Fig, top panel). Germination involves release of dipicolinic acid (DPA) from the spore core. DPA accumulated and/or was retained by the spores can also influence germination and thus the impaired germination could result from reduced extrusion of DPA. To test this point, we measured the DPA content of the various spores. We found the DPA content, normalized to the WT, of Δ*yabG* (125.5% ± 11) and *yabG*$^{C207A}$ (118.8% ± 10) spores to be not significantly different than that of WT (100%) and *yabG*$^C$ (116.7% ± 2) spores (S8C Fig, middle panel). Moreover, WT spores released 0.4 ± 0.04 DPA (expressed as a ratio of the $OD_{270}/OD_{600}$) during germination, *yabG*$^C$ spores released 0.3 ± 0.02 DPA, while Δ*yabG* and *yabG*$^{C207A}$ spores released respectively 0.3 ± 0.01 and 0.4 ± 0.09 DPA (S8C Fig, bottom panel). Thus, 1 hour after induction of germination, the DPA released from *yabG* mutant spores is not significantly reduced compared to WT and *yabG*$^C$ spores.

Only CspB (55 kDa) was detected in extracts from WT or *yabG*$^C$ spores; in contrast, both CspBA and CspB* (see above) were detected in Δ*yabG* and *yabG*$^{C207A}$ spores (S8D Fig) [42]. Pre-pro-SleC (47 kDa) was processed to its Pro form (34 kDa) in WT and *yabG*$^C$ spores, but only full-length SleC was detected in Δ*yabG* or *yabG*$^{C207A}$ spores (S8D Fig, middle panel). Finally, the levels of CspC did not differ significantly between WT and Δ*yabG* spores or between *yabG*$^C$ and *yabG*$^{C207A}$ spores (S8D Fig, bottom panel). Thus, the partial germination defect of Δ*yabG* and *yabG*$^{C207A}$ spores may be due, at least in part, to loss of YabG activity, which in turn impairs the release of CspB from CspBA and the production of pro-SleC. Because germination is also influenced by the status of the spore surface layers, we next wanted to investigate a possible role of YabG in the assembly of the coat and exosporium.

## YabG controls the assembly of various coat and exosporium proteins

We first examined the collection of coat/exosporium proteins that could be extracted from purified WT, Δ*yabG*, *yabG*$^C$ and *yabG*$^{C207A}$ spores. Spores of the four strains were purified on gradients of metrizoic acid, and the proteins extracted in a buffer with SDS and reducing agents; the resulting spores were then treated with lysozyme and proteins again extracted (see the Material and Methods). This produced a cortex/coat/exosporium fraction and a core/cortex fraction (as explained below). Proteins in the two fractions were then analysed by SDS-PAGE and immunoblotting. In Coomassie-stained gels, several proteins were more extractable or more abundant in the cortex/coat/exosporium fractions from Δ*yabG* or *yabG*$^{C207A}$ spores (Fig 2A, red arrowheads). Mass spectrometry analysis identified SpoIVA in the band at around 70 kDa (the expected mass of SpoIVA), CotE in a band at around 81 kDa (close to the expected mass of the full-length protein, 81.3 kDa) (Fig 2A, red arrowheads) [23,53,54]. Also, the abundance or extractability of a form of CdeC was also slightly increased in the Δ*yabG* mutant relative to the WT (Fig 2A, band close to the 50 kDa marker in the

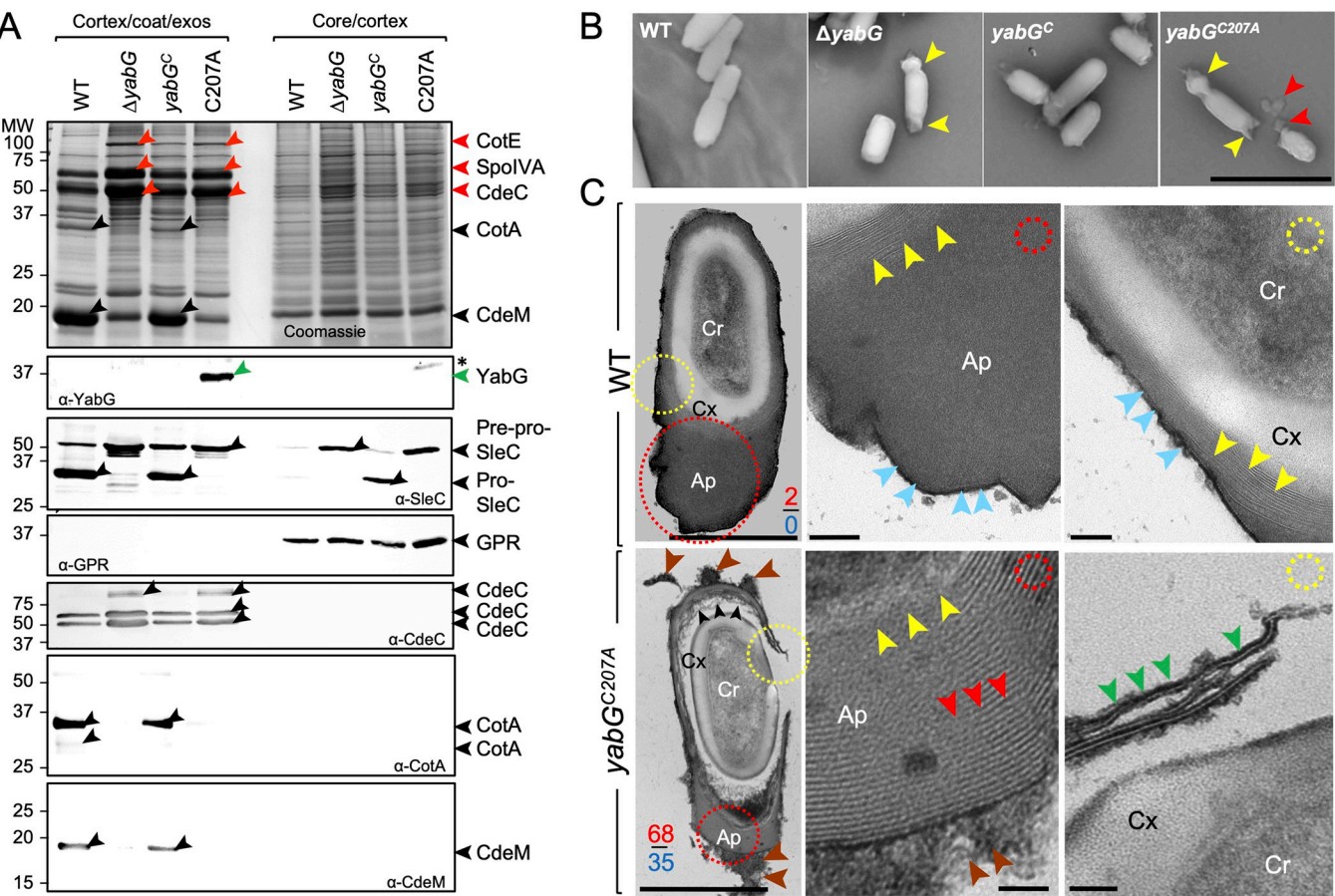

**Fig 2. *yabG* mutations affect the extractability of coat and exosporium proteins and result in abnormal spore surface layers. A**: Spores of the WT, Δ*yabG*, *yabG*^C (complementation strain) and *yabG*^C207A strains were purified on density gradients, fractionated [107] and the cortex/coat/exosporium and cortex/core proteins extracted. The proteins were resolved by SDS-PAGE and the gels stained with Coomassie (top panel) or subjected to immunoblot analysis (lower panels) with anti-YabG, anti-SleC, anti-GPR, anti-CdeC, anti-CotA and anti-CotM antibodies, as indicated. The red arrowheads indicate proteins that appear to be more extractable from Δ*yabG* and *yabG*^C207A spores while black arrowheads show proteins with reduced extractability. The position of YabG^C207A is indicated by a green arrowhead; asterisks show the position of non-specific species. Proteins in the indicated bands in the Coomassie-stained gel were identified by mass spectrometry. **B**: Spores of the indicated strains imaged by scanning electron microscopy. The yellow arrowheads point to the polar regions in *yabG* and *yabG*^C207A spores. The red arrowhead points to material that seems to peel-off the pole of a *yabG*^C207A spore. **C**: Spores of the WT (630Δ*erm*) and *yabG*^C207A mutant were analysed by thin sectioning transmission electron microscopy (TEM). The regions within the red and yellow circles in the two left panels are magnified on the right panels. In WT spores, the yellow arrowheads point to the coat region and the blue arrowheads to the exosporium region. In *yabG*^C207A spores, the red arrowheads point to the lamellae seen in the appendage region, the green arrowheads to electron dense coat or exosporium material loosely attached to *yabG*^C207A spores and the black arrowheads to unstructured material present between the cortex and the coat layers. Also note the material peeling off from the appendage (brown arrowheads). Cr, spore core; Cx, cortex; Ap, appendage region. The numbers refer to the percentage of spores in which the coat is detached from the cortex (red) or with a long polar appendage with a lamellar structure (blue); 60–95 spores were counted for each strain. Scale bars: 3 μm in **B**; 500 nm (left column) or 100 nm (all other panels) in **C**. In **B** and **C**, the spores were purified on density gradients. See also S12 Fig.

Coomassie-stained gel and the immunoblot). Also note that for the Δ*yabG* and *yabG*^C207A mutants, a form of CdeC that migrates above the 75 kDa marker is also more abundant or extractible as detected by immunoblotting (Fig 2A). The increased extractability of SpoIVA from Δ*yabG* spores has been observed before [42]. CotE, found in both the coat and exosporium, is a bi-functional enzyme with a peroxiredoxin N-terminal domain and a C-terminal chitinase domain [23,79,80]. CotE is detected in extracts from WT spores as species of 81 kDa, 40 kDa and a minor species of 38 kDa [23,79]. The accumulation of the 81 kDa species in the *yabG* mutants, suggests that YabG is required for CotE interdomain processing through which the peroxiredoxin and chitinase moieties are separated. In contrast, CdeM and CotA, were

absent or much less extractable from both Δ*yabG* or *yabG^{C207A}* spores as assessed by Coomassie staining (Fig 2A). CdeM, a major component of the exosporium of *C. difficile* spores, is detected by immunoblotting in the cortex/coat/exosporium fraction from WT and *yabG^C* spores as an abundant species at around 17 kDa [24,25] (Fig 2A). CotA is a surface exposed protein required for the assembly of the outer spore surface layers [79,80]. The protein was detected in the cortex/coat/exosporium fraction of WT and *yabG^C* spores as a main species in the 34 kDa region of the gel, consistent with its predicted mass (34 kDa) but also as a species just above the 25 kDa marker (Fig 2A); both forms were previously observed [80]. CotA and CdeM remained undetected in the core/cortex fraction (Fig 2A). YabG itself (32 kDa) could only be detected in the cortex/coat/exosporium extracts prepared from *yabG^{C207A}* spores (Fig 2A, green arrowhead) presumably because the auto-proteolytic activity of YabG^{WT} limits its accumulation. Alternatively, YabG^{WT} may be much less extractable than YabG^{C207A} (see also below). As expected, the spore core protein GPR was detected by immunoblotting only in the core/cortex fraction (Fig 2A). SleC, however, shown before to be associated with the cortex [74] was detected in the cortex/coat/exosporium fraction of WT or *yabG^C* spores, mostly in its processed form (Fig 2A). Trace amounts of Pre-pro-SleC and not the processed form were detected in the core/cortex fraction of WT spores; this may reflect different extractability of the protein from spores of 630Δ*erm* relative to strain R20291 [74]. However, the processed form was clearly visible in this fraction of *yabG^C* spores (Fig 2A). Only the Pre-pro-SleC protein was detected in *yabG* mutants, in either fraction (Fig 2A). While enriching for coat/exosporium proteins, which are not detected in the core/cortex fraction, our extraction procedure also releases cortex-associated proteins, at least SleC. That proteins known to be associated with the cortex are extracted with the coat/exosporium has been reported before [81].

In all, the Δ*yabG* or *yabG^{C207A}* mutations affected the extractability of proteins important for coat (SpoIVA, CotE, CotA) and exosporium morphogenesis (CdeM) and processing of Pre-pro-SleC.

## *yabG* spores are more permeable to lysozyme

Since the integrity of the surface layers is important to prevent access of peptidoglycan-breaking enzymes to the spore cortex, we tested the resistance of spores to lysozyme treatment. Density-gradient purified spores were plated in a rich medium in the presence of the germinant taurocholate, before or after treatment with lysozyme. Survival was of 98.3% for WT spores, 69% for Δ*yabG* spores 72.3% for *yabG^{C207A}* spores and 90.6% for *yabG^C* spores. These numbers are close to the efficiency of plating of spores onto plates containing taurocholate without lysozyme treatment (see above) and were not statistically significant as assessed by one-way ANOVA and Tukey's multiple comparison tests, suggesting that lysozyme had no effect on spore survival. Nevertheless, upon exposure to lysozyme, 46% of the Δ*yabG* spores and 41% of the *yabG^{C207A}* spores become phase-dark (6% for WT spores and 15% of the *yabG^C* spores) (S10 Fig). Thus, the alterations in assembly of the coat/exosporium layers appear sufficient to allow access of lysozyme to the spore cortex.

## YabG is required for proper spore morphogenesis

Because spores formed by the Δ*yabG* and *yabG^{C207A}* mutants show alterations in the levels or extractability of several coat/exosporium proteins and are more permeable to lysozyme, we characterized the morphology of the spores produced by the two mutants. Purified spores were first imaged using phase contrast microscopy. Spores of *C. difficile* often possess an appendage at one of the spore poles, which is continuous with the exosporium [25]. About 12% of spores produced by the WT (15% for the *yabG^C* strain) possess a well-developed polar

appendage, a number close to that (20%) previously reported [25] (S11A and S11B Fig). Strikingly, the Δ*yabG* or *yabG*<sup>C207A</sup> alleles increased the fraction of spores with a visible appendage to 26% and 24%, respectively (S11A and S11B Fig). Two features, however, distinguished the appendage region in WT or *yabG* spores: firstly, in the Δ*yabG* or *yabG*<sup>C207A</sup> mutants, the appendage tended to be more square in shape (S11A Fig, red arrowheads); secondly, while in WT spores, the appendage does not stain with FM4-64, it does so in the two mutants (S11C Fig). Consistent with this result, the fluorescence intensity signal for FM4-64 in the spore polar appendage region, is higher in spores of *yabG* mutants (S11D Fig). Furthermore the intensity signal for FM4-64 in *yabG*<sup>C207A</sup> spores was higher than in Δ*yabG* spores(S11D Fig), indicating a possible difference in the composition and/or morphology between *yabG*<sup>C207A</sup> and Δ*yabG* spores.

Prominent polar appendages, occasionally present at both spore poles, were also seen by scanning electron microscopy (SEM) (Fig 2B, yellow arrowheads). Moreover, material often appeared to detach from this region of the spore, suggesting misassembly of the appendage (Fig 2B, red arrowheads). To characterize the ultrastructure of spores, we used thin sectioning transmission electron microscopy (TEM). The main features previously documented for spores of the WT strain 630Δ*erm* were observed, including the compact core, the cortex, and the lamellar coat surrounded by a thin, more electron dense, exosporium (Fig 2C, blue arrowheads). The appendage itself shows a compact organization and has a well-defined coat/appendage transition zone at its base [25]; (Fig 2C, yellow arrowheads). Strikingly, the structure of the appendage is markedly different in the mutants, consistent with the FM4-64 staining results (above). The transition zone at the basis of the appendage is less defined (Figs 2C and S12, yellow arrowheads) and consistent with the SEM analysis (above), coat/exosporium material seems to peel of the spore poles in both *yabG*<sup>C207A</sup> (Fig 2C, brown arrowheads) and Δ*yabG* spores (S12 Fig.). Importantly, in about 35% of the *yabG*<sup>C207A</sup> spores with appendages, and in 32% of the Δ*yabG* spores, the appendage appears as a series of closely juxtaposed lamellae (Fig 2C, red arrowheads in the lower set of panels and S12 Fig). This feature is clearly distinct from the compact appearance of the appendage in WT or *yabG*<sup>C</sup> spores (Figs 2C and S12). Only 2% of the WT spores and 9% of the *yabG*<sup>C</sup> spores showed the coat detached from the cortex, and no spores with a lamellar structure of the appendage could be seen for either strain (Figs 2C and S12). Nevertheless, In WT or *yabG*<sup>C</sup> spores, hints of a lamellar organization are occasionally seen at the edges of the appendage (Fig 2C and S12) [81,82]. Possibly, the appendage has an underlying lamellar structure upon which proteins such as CdeM, are deposited to form the compact, electron dense appendage. In Δ*yabG* or *yabG*<sup>C207A</sup> spores, the absence of these proteins reveals the underlying structural organization of the appendage, which may be shared by the rest of the exosporium. Consistent with the absence of CdeM from the extracts of *yabG* spores (above, [24]), the exosporium-like layer is often absent in spores of the Δ*yabG* or *yabG*<sup>C207A</sup> spores (Figs 2C and S12). Also, in 68% of the spores of a *yabG*<sup>C207A</sup> mutant, and in 85% of the Δ*yabG* spores, the coat/exosporium did not adhere to the underlying cortex (Fig 2C, yellow circle in the lower set of panels; see also S12 Fig) and disorganized material was seen in the region between the cortex and the coat/exosporium (Fig 2C, black arrowheads in the bottom set of panels; S12 Fig). The impaired adherence of the coat/exosporium to the cortex may explain why in the phase contrast images, the percentage of spores with a recognizable appendage increases in the *yabG* mutants, as they are easier to recognize (S12 Fig). Finally, in Δ*yabG* and *yabG*<sup>C207A</sup> spores, the coat/exosporium shows regions of discontinuity along the periphery of the spore (Fig 2C, region within the yellow circle in the lower set of panels, and green arrowheads in the magnified image). These discontinuities likely contribute to the increased accessibility of lysozyme to the cortex layer in *yabG* spores.

## YabG is required for the expression of *cotA* and *cdeM*, involved in coat and exosporium assembly

One explanation for the absence of CotA and CdeM from the coat/exosporium extracts of *yabG* spores, is that somehow YabG could affect production of these proteins. To test this possibility, we first analysed the accumulation of CotA, CdeM and CdeC, for reference, by immunoblotting, in whole cell extracts prepared from sporulating cells harvested after 14 and 20 hours of growth on 70:30 agar plates. CotA and CdeM were detected in the extracts prepared from the WT at 14 and 20 hours of growth but not in the extracts prepared from the *yabG* mutants; in contrast, CdeC was detected in the WT and the mutants at both time points (Fig 3A and 3B). Thus, the absence of CdeM and CotA from the coat/exosporium extracts of *yabG* and *yabG^C207A* spores appears to be a consequence of reduced synthesis or accumulation of the two proteins.

We next measured the transcript levels of *cotA*, *cdeM* and *cdeC*, in sporulating cells of the WT and the Δ*yabG* mutant using qRT-PCR. These experiments showed decreased levels of *cotA* and *cdeM* transcripts in Δ*yabG* cells compared to the WT both at 14 h (expression ratio Δ*yabG*/WT of 0.060 for *cotA* and 0.216 for *cdeM*) and at 20h of growth (expression ratio Δ*yabG*/WT of 0.098 and 0.126, respectively) (Fig 3C). In contrast, *cdeC* expression increased from a Δ*yabG*/WT ratio of 1.873 at 14 hours to 7.139 at hour 20 (Fig 3C). While not excluding a direct role for YabG in the assembly of CotA and CdeM (but see section below), these results show that *yabG* is required for the expression of *cotA* and *cdeM*. Moreover, since both the Δ*yabG* and *yabG^C207A* mutations strongly reduce the levels of the two proteins in spores, we infer that the proteolytic activity of YabG is required for the expression of *cotA* and *cdeM*. One possibility is that the activity of YabG is required for the removal of a negative regulator of *cotA* and *cdeM* expression; the increased expression rate of *cdeC* observed at 20 h of growth might be an indirect effect of the absence of YabG activity at late stages of sporulation (see also the Discussion).

## Bypass of YabG for expression of *cotA* and *cdeM* and assembly of CdeM and CotA

That both the Δ*yabG* and *yabG^C207A* alleles reduced the levels of the *cotA* and *cdeM* transcripts raised the possibility that the activity of YabG is somehow required to antagonize a transcriptional repressor or to activate a factor required for transcription of both genes. If so, then replacing the *cotA* and *cdeM* promoters by a *yabG*-independent promoter, should bypass the need for *yabG* for CotA and CdeM production. Both genes were placed under the control of the *cotE* promotor; *cotE* also codes for a late coat protein, produced under the control of σ^K; the *cotE* promoter, however, does not appear to be YabG-dependent, since CotE^FL accumulates in *yabG* spores (Fig 2A). Strains bearing the P*_cotE_*-*cotA* and P*_cotE_*-*cdeM* fusions in the WT, *yabG* and *yabG^C207A* backgrounds were grown under sporulation conditions, spores were purified and coat/exosporium extracts prepared and analysed. CdeM is was detected in Coomassie-stained gels [25], whereas the presence/absence of CotA in spores extracts required verification by immunoblotting. Expression of *cdeM* or *cotA* from the *cotE* promotor restored the presence of CdeM (Fig 4A) and CotA (Fig 4B, bottom panel) in extracts from Δ*yabG* or *yabG^C207A* spores.

To determine if and to what extent expression of *cdeM* or *cotA* corrected the phenotype of *yabG^C207A* spores we used TEM. In spores of the *yabG^C207A* mutant expressing of P*_cotE_*-*cdeM* the lamellar appearance of the polar appendage region was lost (Fig 4C, middle panel); instead, the appendage region appeared compact and electrondense (Fig 4C, Ap), consistent with the accumulation of CdeM and its role in formation of the polar appendage [25]. Other features of

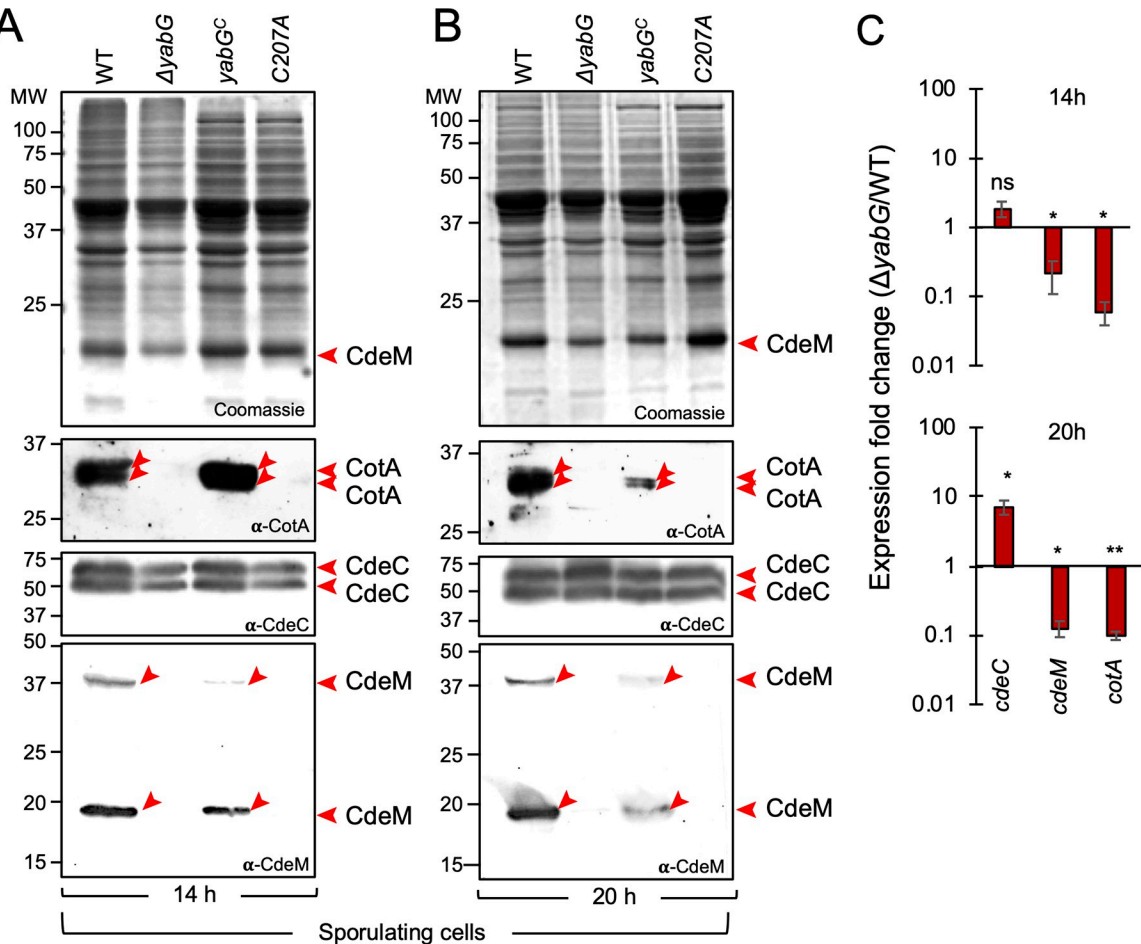

**Fig 3. YabG affects the expression of genes coding for coat and exosporium components. A and B:** Coomassie stained gel and immunoblotting analysis of sporulating cells of the WT, Δ*yabG*, *yabG^C* and *yabG^{C207A}* 14 (**A**) and 20 hours (**B**) after inoculation in 70:30 agar plates [53]. Proteins in whole cell extracts were resolved by SDS-PAGE and the gels subjected to immunoblotting with anti-CotA, anti-CdeC, and anti-CdeM antibodies. The red arrowheads point to the various forms of CotA, CdeC and CdeM. **C:** Quantification of the expression of the indicated genes (*cdeC*, *cdeM* and *cotA*) by qRT-PCR. Total RNA was extracted from *C. difficile* 630Δ*erm* and Δ*yabG* strains grown in 70:30 agar plates for 14 and 20 hours. The graph shows the fold-change in the expression of *cdeC*, *cdeM* and *cotA* between the Δ*yabG* and the WT. Error bars correspond to the standard deviation derived from three biological replicates. Statistical analysis used a Student's t-test: * p<0.01; **p<0.001.

*yabG* spores, however, such as the peeling off of significant sections of the coat/exosporium were maintained (Fig 4C, blue arrowheads). Expression of P*cotE*-*cotA* in the *yabG^{C207A}* background resulted in spores with visible juxtaposed sheets in the polar appendage region (Fig 4C, panels in the right), similar to *yabG* spores and most likely due to the absence of CdeM (see also above).

These results show that YabG acts at the level of the *cotA* and *cdeM* promoters to influence transcription of these genes. In addition, since the expression of *cdeM* and *cotA* from P*cotE* result in the detection of CdeM and CotA in coat/exosporium extracts prepared from *yabG* spores, we infer that YabG is not a strict requirement for the localization of CotA or CdeM.

## Auto-regulatory assembly of YabG

We then wanted to monitor the sub-cellular localization of YabG. To this end, a translational fusion of the WT protein to the SNAP^Cd tag, YabG^WT-SNAP^Cd, was constructed and

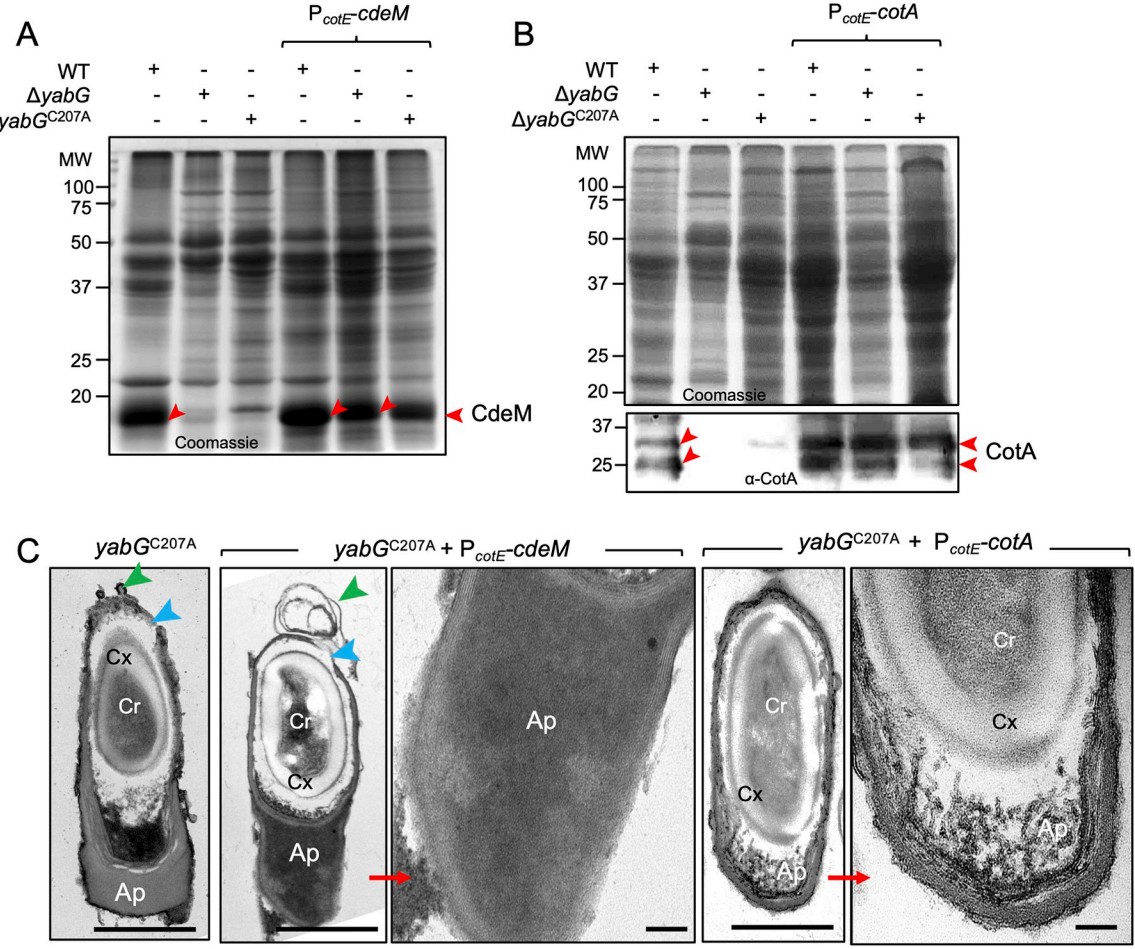

**Fig 4. YabG-independent expression of *cotA* and *cdeM* restores assembly of CotA and CdeM to Δ*yabG* and *yabG*^C207A spores.**
Coomassie stained SDS-PAGE gel of the proteins extracted from purified spores of the WT, Δ*yabG* and *yabG*^C207A mutants and
derivatives expressing P*cotE*- *cdeM* (**A**) or P*cotE*- *cotA* (**B**). In **A**, the bottom panel shows the immunoblot analysis of the corresponding gel
using an anti-CotA antibody. The position of the main forms of CdeM (in **A**) or CotA (in **B**) is shown by red arrowheads. **C**: spores
produced by the *yabG*^C207A mutant and by derivatives expressing P*cotE*-*cdeM* or P*cotE*-*cotA* were imaged by TEM. The polar region of
*yabG*^C207A/P*cotE*-*cotA* and *yabG*^C207A/P*cotE*-*cdeM* spores is magnified in the panels to the right. Cr, core; Cx, cortex; Ap, spore appendage
region. Green arrowheads, coat and exosporium material peeling off the spore; blue arrowheads, regions with an exposed cortex. Scale
bars: 100 nm for the magnified images, 500 nm for all other panels.

introduced into the WT strain. As detailed below, the YabG^WT-SNAP^Cd fusion is largely func-
tional. Cells were grown in 70:30 agar plates and imaged by phase contrast and fluorescence
microscopy 14 and 20 hours after inoculation. At hour 20 in the WT background, *i.e.*, in the
presence of the *yabG*^WT allele, YabG^WT-SNAP^Cd was detected around 10% of the phase-dark
forespores, in 42% of the phase-grey forespores and in 81% of the phase-bright forespores (Fig
5A, yellow arrowheads). A similar localization pattern was observed for YabG^WT-SNAP^Cd at
hour 14 (S13A Fig, yellow arrowheads on the left set of panels) and at the two time points in a
Δ*yabG* mutant (Figs 5A and S13A). Thus, consistent with an association of the protein with
the coat and/or exosporium layers YabG^WT-SNAP^Cd localizes to the forespore after engulf-
ment completion and remains associated with the developing spore at late stages in morpho-
genesis. We note that for both YabG^WT-SNAP^Cd and YabG^C207A-SNAP^Cd, in either the WT or
Δ*yabG* background, a haze of fluorescence is detected in the mother cell cytoplasm (Figs 5A
and S13A, white arrowheads). This signal may result from release of the SNAP^Cd moiety

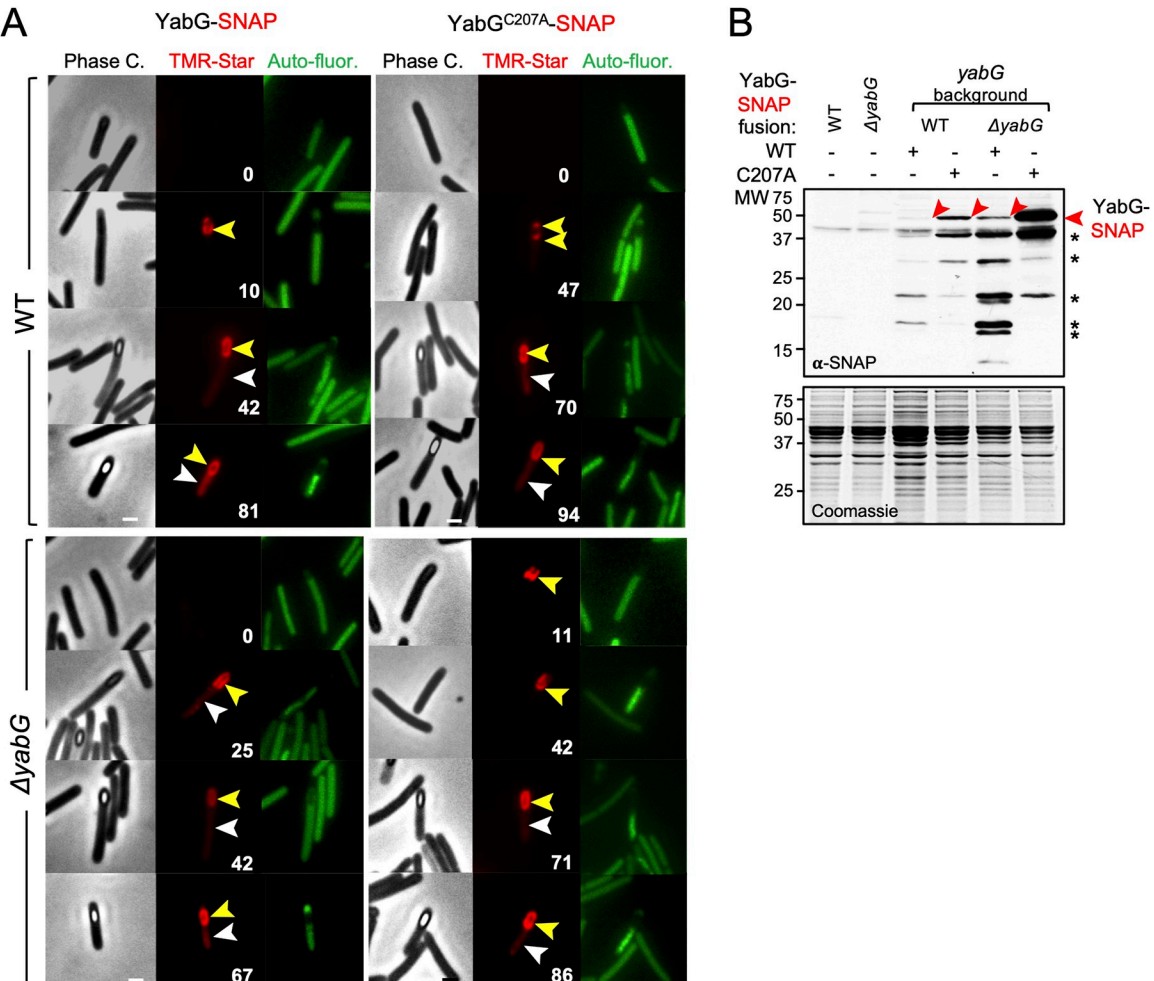

**Fig 5. Localization of YabG-SNAP$^{Cd}$ in sporulating cells. A:** Localization of YabG$^{WT}$-SNAP$^{Cd}$ and YabG$^{C207A}$-SNAP$^{Cd}$ in *C. difficile* 630Δ*erm* (WT) and Δ*yabG* strains. Cells were collected after 20h of growth in 70:30 agar plates [53], stained with the SNAP substrate TMR-Star and examined by phase contrast and fluorescence microscopy (red channel for TMR signal and green channel for autofluorescence signal). The numbers refer to the percentage of cells at the represented stage showing SNAP fluorescence. Yellow and white arrowheads point to the position of the forespore and the mother cell respectively. At least 150 cells were analysed for each strain, in three independent experiments. Scale bar, 1 μm. **B:** Accumulation of YabG-SNAP$^{Cd}$ and YabG$^{C207A}$-SNAP$^{Cd}$ in sporulating cells of strains 630Δ*erm* (WT) and Δ*yabG* at 20h of growth in 70:30. Proteins in whole cell extracts were resolved by SDS-PAGE and the gel subject to immunoblot analysis with anti-SNAP antibodies. Samples collected from the WT and the Δ*yabG* mutant bearing no SNAP$^{Cd}$ fusion were used to control for antibody specificity. The Coomassie-stained gel is included as a loading control. Red arrowheads point to the position of YabG$^{C207A}$-SNAP (52 kDa). Asterisks denote possible degradation products that include the SNAP moiety (~19.4 kDa). The black arrowhead shows the position of a cross-reactive species.

through proteolysis or otherwise indicate that some of the fusion proteins remain in the mother cell (see below).

To determine whether the activity of YabG was involved in the association of the protein with the developing spore, we monitored the localization of the catalytically-inactive YabG$^{C207A}$-SNAP$^{Cd}$ fusion. At 20 hours of growth, YabG$^{C207A}$-SNAP$^{Cd}$ localized as 2 caps at the mother cell proximal and distal poles in 47% of phase-dark spores (Fig 5A). YabG$^{C207A}$-SNAP$^{Cd}$ formed a ring of fluorescence around 70% of phase-grey forespores and around 94% of phase bright-forespores (Fig 5A, yellow arrowheads). A similar pattern of localization was observed at hour 14, except that YabG$^{C207A}$-SNAP$^{Cd}$ was detected even earlier, as a single cap of fluorescence in 6% of the sporangia during engulfment (S13A Fig., yellow arrowheads). The

localization of YabG$^{C207A}$-SNAP$^{Cd}$ in cells during engulfment and the higher percentages of localization of the fusion protein in sporangia of phase-dark, phase-grey and phase-bright spores suggests increased stability of the catalytically inactive protein even in the presence of the WT *yabG* allele. This observation suggests that the auto-proteolytic activity of YabG, detected for both the *B. subtilis* [52] and the *C. difficile* proteins (Fig 1E) controls the accumulation and localization of the protein. If so, the localization of the fusion proteins could increase in cells of a Δ*yabG* mutant. In comparison to the WT background, however, the localization of YabG$^{WT}$-SNAP$^{Cd}$ in Δ*yabG* sporangia only increased slightly around phase-dark forespores (25% as opposed to 10% in the WT at hour 20; Fig 5A) and for phase-grey forespores (65% as opposed to 55% at hour 14; S13A Fig). For the localization of YabG$^{C207A}$-SNAP$^{Cd}$ in the Δ*yabG* background, the main difference relative to the WT background was the increase in the single cap pattern in sporangia during engulfment (from 0 to 11% at hour 20 and from 6 to 12% at hour 14; Figs 5A and S13A). The immunoblot analysis of whole cell extracts is in good agreement with the microscopy results. YabG$^{C207A}$-SNAP$^{Cd}$ is detected with an anti-SNAP monoclonal antibody at higher levels than YabG$^{WT}$-SNAP$^{Cd}$ (both proteins run as 52 kDa species) in both the WT and the Δ*yabG* background with the highest accumulation corresponding to the catalytically inactive fusion in the Δ*yabG* mutant (red arrowheads in Figs 5B and S13B.). Bands just above and below the 20 kDa marker are likely to result from cleavage of the fusion protein close to the C-terminus of YabG (asterisks in Figs 5B and S13). The increased accumulation of YabG$^{WT}$-SNAP$^{Cd}$ or YabG$^{C207A}$-SNAP$^{Cd}$ in *yabG* mutants does not seem to result from augmented transcription of *yabG*: control experiments show that the transcription of a P$_{yabG}$-SNAP$^{Cd}$ fusion increases only slightly in Δ*yabG* sporangia (S14 Fig).

Thus, YabG$^{WT}$-SNAP$^{Cd}$ appears to degrade itself; since no major difference was detected for YabG$^{C207A}$-SNAP$^{Cd}$ in Δ*yabG* sporangia in comparison to the WT, it may be that the degradation of YabG requires at least one *in cis* cleavage event. In any event, since YabG$^{C207A}$-SNAP$^{Cd}$ localizes earlier than the WT, the catalytic activity of YabG controls, at least in part, the localization of the protein during sporulation. In that sense, assembly of YabG is auto-regulatory.

## YabG localizes asymmetrically in mature spores

To gain insight onto the localization of YabG in mature spores, we used Super Resolution Structured Illumination microscopy (SR-SIM), in which the lateral resolution is increased to about 110 nm, as compared with the diffraction limit of 250 nm of conventional light microscopy [83]. Spores of strains producing either YabG$^{WT}$- or YabG$^{C207A}$-SNAP$^{Cd}$ in the WT or Δ*yabG* backgrounds were labelled with MTG, which decorates the spore body and with TMR-Star prior to SR-SIM imaging. Both YabG$^{WT}$- and YabG$^{C207A}$-SNAP$^{Cd}$ localized around the entire contour of the spore (Fig 6A, white arrowheads; see also S15 Fig), in confirmation of the conventional fluorescence microscopy data where the signal for YabG-SNAP was detected around the forespore in sporangia of phase-bright forespores (above). Strikingly, in the WT, YabG$^{WT}$-SNAP$^{Cd}$ localized to and showed a strong signal overlapping the spore polar appendage in 63% of the spores showing this structure, and YabG$^{C207A}$-SNAP$^{Cd}$ localized in 85% of those spores (Fig 6A, blue arrowheads; see also S15 Fig). In the Δ*yabG* mutant, YabG$^{WT}$-SNAP$^{Cd}$ localized in 63% of the appendage-bearing spores, and YabG$^{C207A}$-SNAP$^{Cd}$ localized in 46% of those spores (Fig 6A, blue arrowheads; S15 Fig). The impaired localization of the YabG$^{C207A}$ fusion in the Δ*yabG* background suggests that the WT protein facilitates the localization of the mutant form to free spores and/or its maintenance. The localization of the fusion proteins to the spore polar appendage is consistent with the role of YabG in the morphogenesis of this structure (see above).

Proteins in a cortex/coat/exosporium and a core/cortex fraction were resolved by SDS-PAGE and analysed by immunoblotting. Full-length YabG$^{WT}$-SNAP$^{Cd}$, with an expected

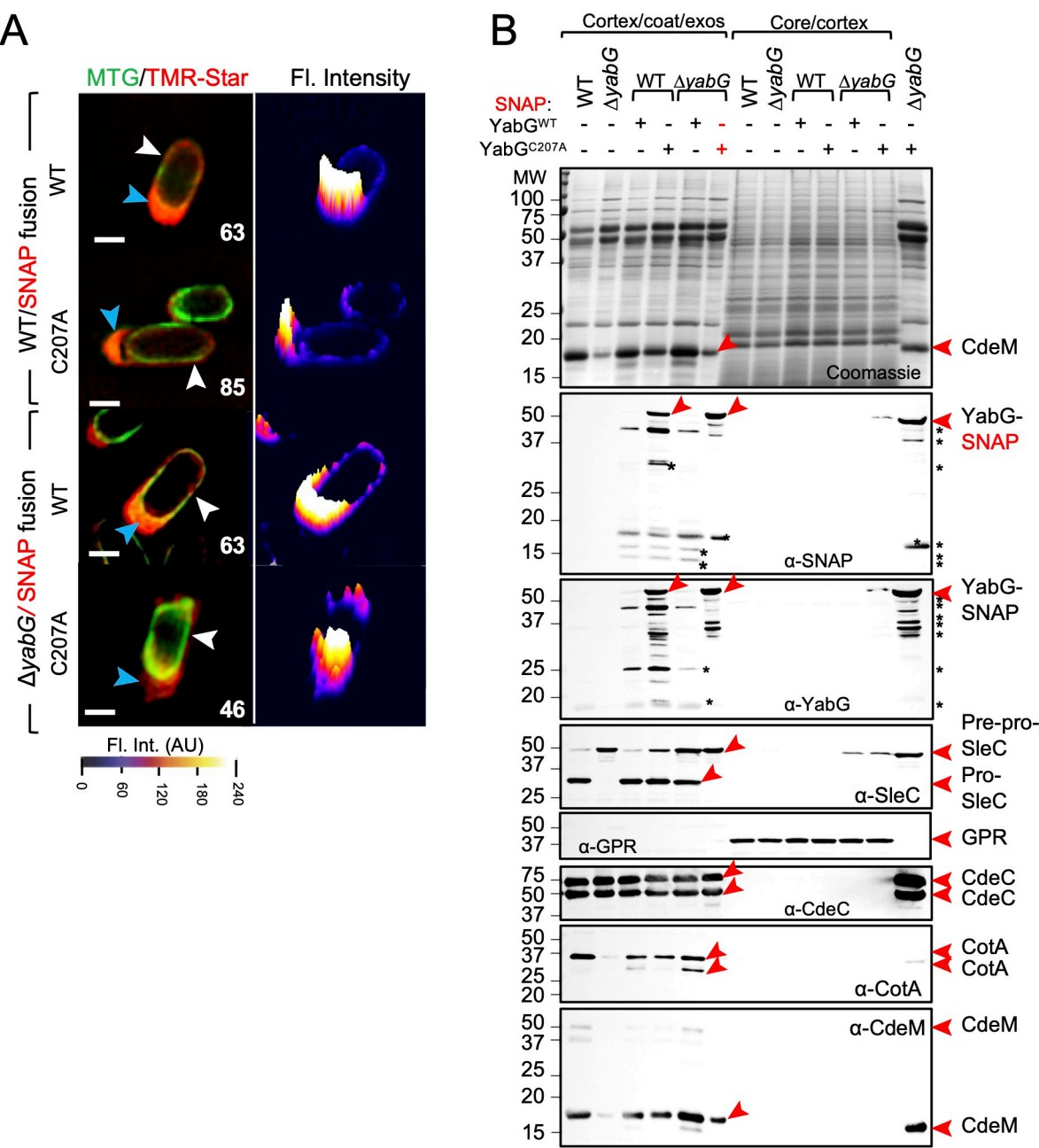

**Fig 6. Localization of YabG^WT- or YabG^C207A-SNAP^Cd in mature spores. A:** Localization of YabG^WT- or YabG^C207A-SNAP^Cd in mature spores using SR-SIM, in either the WT or Δ*yabG* backgrounds. The spores were stained with the membrane dye MTG (green) and with TMR-Star (red) prior to imaging. The blue arrows point to the SNAP signal at the spore poles and the white arrows to the signal along the side of the spore (see also S15 Fig). The distribution of the fluorescence signal (in arbitrary units, AU) in three dimensional intensity graphs is shown below the microscopy images. Scale bar, 500 nm. **B:** Coomassie stained SDS-PAGE gel of the proteins extracted from the cortex/coat/exosporium and core/cortex fractions of purified spores of the WT strain, the Δ*yabG* mutant, and of strains producing YabG^WT-SNAP^Cd or YabG^C207A-SNAP^Cd in either the WT or Δ*yabG* backgrounds. The gel was subjected to immunoblot analysis with anti-SNAP, anti-YabG, anti-SleC, anti-CdeC, anti-CotA and anti-CdeM antibodies. The arrowheads points to the position of the relevant proteins. YabG-SNAP denotes the position of either full-length WT or the C207A variant fused to the SNAP tag. Asterisks denote possible degradation products or cross-reactive species.

size of about 52 kDa, was barely detected in the coat/exosporium fraction of WT or Δ*yabG* spores with anti-SNAP or anti-YabG antibodies and was not detected in the core/cortex fraction of either strain (Fig 6B, red arrowhead). In contrast, YabG$^{C207A}$-SNAP$^{Cd}$ was detected in the cortex/coat/exosporium fraction of both WT and Δ*yabG* spores, and in trace amounts in the core/cortex fraction of Δ*yabG* spores (Fig 6B). Since full-length YabG$^{WT}$-SNAP$^{Cd}$ does not accumulate even in the Δ*yabG* mutant, it seems that the protein undergoes auto-proteolysis and thus, that the fusion protein is largely functional with respect to this activity (see also below). Auto-proteolysis may occur, at least in part, *in cis*, since full-length YabG$^{C207A}$-SNAP$^{Cd}$ was detected in the WT but at significantly higher levels in the Δ*yabG* mutant (Fig 6B, middle panel, red arrowhead). Because the fluorescence signal from YabG$^{WT}$- or YabG$^{C207A}$-SNAP$^{Cd}$ in the SR-SIM images is comparable (Fig 6A), we infer that YabG$^{WT}$-SNAP$^{Cd}$, although accessible to the TMR-Star substrate, is less extractable than YabG$^{C207A}$-SNAP$^{Cd}$.

Bands below the 20 kDa marker, detected in the WT for YabG$^{WT}$-SNAP$^{Cd}$ and for YabG$^{C207A}$-SNAP$^{Cd}$, may result from cleavage of the fusion to release the SNAP moiety or fragments of it, since the C-terminally located reporter has a predicted molecular mass of 19.4 kDa (Fig 6B). As above (Fig 2A), SleC was only detected in the cortex/coat/exosporium fraction [74], whereas GPR was only detected in the core/cortex fraction (Fig 6B). Importantly, the analysis of the cortex/coat/exosporium extracts by Coomassie staining and immunoblotting showed that the YabG$^{WT}$-SNAP$^{Cd}$ fusion, but not YabG$^{C207A}$-SNAP$^{Cd}$, restored both CotA and CdeM assembly and significant processing of Pre-pro-SleC to spores of a Δ*yabG* mutant and no other major differences to WT spores were noticed (Fig 6B). We infer that with respect to the assembly of the spore surface, YabG$^{WT}$-SNAP$^{Cd}$ is largely functional.

## YabG is required for colonization in a hamster model

We then wanted to test whether YabG played a role in colonization or infection in an hamster model. To induce susceptibility to *C. difficile* infection, female Syrian golden hamsters, individually housed in cages were first gavaged with clindamycin [84,85] and after 5 days, gavaged with $10^3$ WT or Δ*yabG* spores. Signs of disease (lethargy, poor fur coat and wet tail) were monitored along time. In parallel, fecal samples were collected daily, plated onto TCCFA plates to enumerate both *C. difficile* vegetative cells and spores (see also the Material and Methods section). The percentage of hamsters that survived 96 hours post infection was similar for the WT and Δ*yabG* spores, indicating that the two strains are equally virulent (Fig 7A).

The total CFU/g counts from day 1 to day 6 post-infection, however, were considerably reduced for Δ*yabG* as compared to the WT (Fig 7B). While for the WT, the number of *C. difficile* cells and spores in the fecal material was between $10^6$ and $10^8$ CFU/g at day 1, this number was below $10^4$ CFU/g for the mutant (Fig 7B). From day 1 to day 2 post infection, the number of CFU/g increased for the WT to between $10^7$ to $10^9$ CFU/g and it remained between $10^7$ and $10^8$ CFU/g up to day 6; in contrast, the number of CFU/g remained low, around $10^5$ CFU/g for the Δ*yabG* mutant (Fig 7B). We note that the number of animals infected with Δ*yabG* spores is lower that for the WT because some of those animals were sick and did not produce a fecal sample; for this reason they were not included in the analysis. In any case, *yabG* seems to have an important role during colonization of the hamster colon.

## Discussion

### Domain organization of YabG

An AlphaFold2 model indicates that YabG has an N-terminal domain separated from the catalytic domain by a linker (Fig 1C). The larger, C-terminal domain B most likely has the fold of receiver domains of response regulators such as CheY [67]. The conserved Asp that is the site

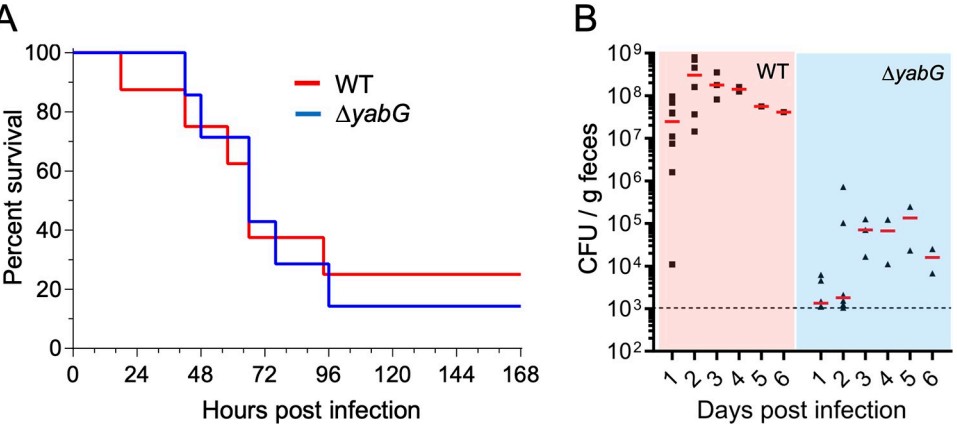

**Fig 7. YabG is required for colonization of the hamster colon. A**: Kaplan-Meier curve for hamster challenged with spores of the WT (red) and Δ*yabG* mutant (blue). Syrian hamsters were first gavaged with clindamycin to induce susceptibility to infection and challenged with 10³ WT or Δ*yabG* spores five days after. Hamsters showing signs of disease were euthanized. **B**: Enumeration of *C. difficile* cells and spores in faecal material following gavage of hamsters with spores produced by the WT (pink panel) and Δ*yabG* mutant (light blue panel) (as in **B**). Faecal samples were collected daily, and both *C. difficile* vegetative cells and spores were enumerated by plating. The data was set to the median values and the limit of detection (dashed line) is shown.

of phosphorylation in CheY and other response regulators, however, is not conserved. Thus, YabG appears to function independently of phosphorylation [68,69]. A significant number of receiver domains, so called single domain response regulators, are not attached to an independently folded output domain and some, such as CheY function by interacting with other proteins [68,69,86]. We do not know whether YabG functions via interactions with proteins other than its substrates, but since the *yabG*[C207A] catalytic inactive mutant phenocopies a *yabG* deletion mutant, this seems unlikely.

As found for the *B. subtilis* protein [52], *C. difficile* YabG also shows an auto-proteolytic activity (Fig 1E). Auto-proteolysis of *B. subtilis* YabG uses a C218/H172 catalytic dyad [52]. These residues are homologous to C207 and H161 of the *C. difficile* protein and may be directly involved in catalysis. Asp162, is also required for the activity of the *C. difficile* protein and is conserved among YabG ortologues but the role of the homologous residue in the *B. subtilis* protein, Asp173, has not been tested. In any event, C207, H161 and D162 are all located in the CheY-like domain B (Fig 1C and 1D). A CheY-like domain showing proteolysis activity is, to our knowledge, unique.

The position of the SH3-like domain A, on top of the putative catalytic center (Fig 1C), suggests that it has to be removed for activation [58]. An alternative view is that the SH3-like domain A is involved in interactions with substrate proteins and is thus required in activity. In favour of this last view, since one of the point mutations found in *yabG* that result in poor processing of Pre-pro-SleC and CspBA and in spores that no longer require co-germinants is located in the A domain [43].

## Substrates of YabG

In *C. difficile*, as also seen in *B. subtilis*, the morphogenetic ATPase SpoIVA is present at higher levels in coat/exosporium extracts prepared from Δ*yabG* spores [38,42]. Thus, SpoIVA may a YabG substrate. None of the other known *B. subtilis* YabG substrates have orthologues in *C. difficile* and conversely, neither CspBA, Pre-pro-SleC, or CotE is found in *B. subtilis* or closely related organisms [18,34,44]. It is likely that at least in *C. difficile*, YabG has additional

substrates. For example, in a *yabG* insertional mutant SleC is activated in response to TA alone, suggesting the involvement of YabG in the processing of an as yet unknown protein presumably controlling germinant specificity [43]. Since YabG is conserved across spore formers, it follows that in spite of a high degree of sequence conservation, YabG must bear structural determinants that allow utilization of specific substrates in different organisms.

### Auto-regulatory assembly of YabG

The catalytically inactive YabG$^{C207A}$ localizes to the forespore surface slightly earlier than the WT protein. Thus, the auto-processing activity translates into delayed assembly of YabG. Localization of YabG$^{WT}$ is first detected when the forespore becomes phase-dark and it increases when the forespore becomes phase-bright, a stage at which expression of *yabG* also increases (S6 and S14 Figs). It thus seems possible that YabG is only assembled when a threshold level of the protein is reached. This threshold may also depend on increased synthesis of its substrates, which may compete for and reduce auto-proteolysis of YabG (Fig 8). Although YabG localizes to the cortex/coat/exosporium its exact location within these structures is not known. SpoIVA is most likely close to the forespore outer membrane and CspB, CspA and Pre-pro-SleC are associated with the cortex possibly in a complex [48,78,87,88]. Possibly, YabG is also located in the innermost layers of the coat.

### Role of YabG in coat/exosporium assembly: Morphological features of yabG spores

YabG has two main effects on the assembly of the spore surface layers. Firstly, it is required for attachment of the coat to the underlying cortex layer (Figs 2C and S12). The basis for this phenotype is unclear. Since SpoIVA localizes close to or at the forespore outer membrane and is required for proper assembly of the cortex and coat [53,54], proper levels and/or processing of this morphogenetic ATPase may be important for coat/cortex attachment. Secondly, YabG is required for assembly of the exosporium. In the absence of CdeM, the exosporium is thin, lacks electrondensity and the appendage is short and disorganized [24,25]. Therefore, the structural defects seen at the level of the exosporium in *yabG* mutants may be explained, at least in part, by the absence of CdeM. Strikingly, in *yabG* spores, the appendage region lacks the compact, electrondense structure seen in WT spores, and instead shows a lamellar organization (Figs 2C and S12). We suspect that the spore polar appendage has an underlying lamellar structure, possibly formed by exosporium proteins such as CdeC, which is revealed in the absence of CdeM. While the localization of YabG around the entire spore is consistent with a role in assembly of the exosporium, its enrichment at the spore poles suggests a specific role in assembly of the polar appendage.

### YabG control of the mother cell line of gene expression

We found that the expression of *cdeM* and *cotA* is severely curtailed in the Δ*yabG* mutant (Fig 3). Expression of *yabG* begins soon after asymmetric division, under the control of σ$^E$ and is maintained during later stages with the (weaker) contribution of σ$^K$ (S6 Fig). In contrast, while *cdeC* expression is first detected after engulfment completion under the direct control of σ$^K$, at the onset of the main period of σ$^K$ activity, expression of at least *cdeM* occurs later, when the forespore turns phase-bright (S14 Fig) [25]. The mechanism by which expression of *cdeM* and possibly other genes is delayed is presently unknown but it could involve YabG. A simple explanation for the effect of YabG on *cotA* and *cdeM* transcription is that the protease is involved in the removal of a factor that represses transcription of some σ$^K$-dependent genes. SpoIIID is an ancillary regulatory protein, produced under the control of σ$^E$, required for the

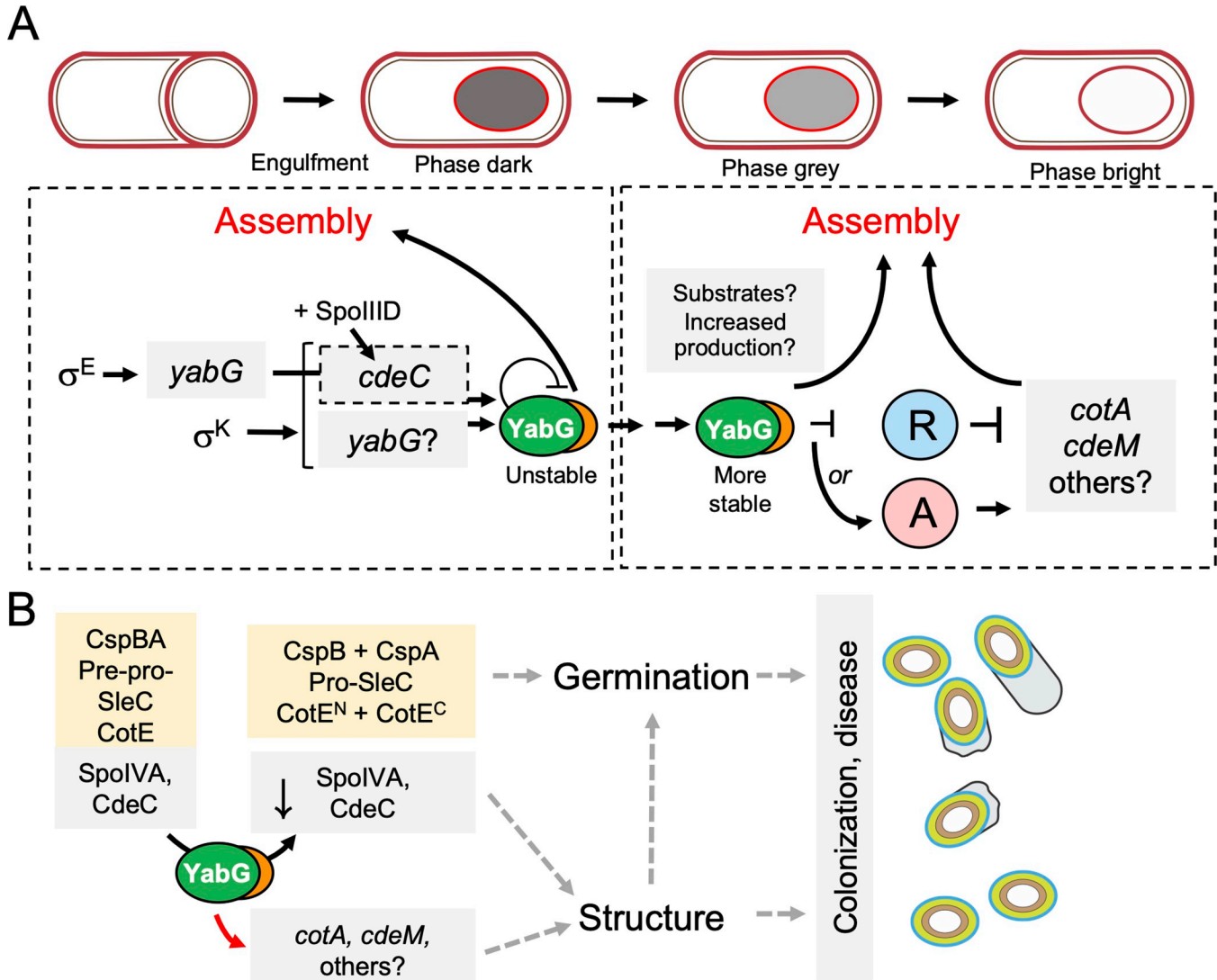

**Fig 8. Model for the role of YabG in the assembly of the spore surface layers. A**: *yabG* expression is first detected in the mother cell compartment after asymmetric division under the control of σ^E but persists in this cell under σ^K control [30]. σ^K also directs transcription of *cdeC* (which additionally requires SpoIIID); transcription of *cdeC* and the σ^K-dependent period of *yabG* transcription begins coincidently with the appearance of phase-dark forespores and CdeC is probably recruited to the forespore surface at this stage. The assembly of YabG, however, is auto-regulatory and self-limiting in that YabG undergoes self-degradation in either the mother cell cytoplasm or at the spore surface. The stability of YabG increases when the spore turns phase-bright, and may require production of the YabG substrates and/or increased production of the protease. At this stage, YabG accumulates at the spore surface and is also required in the mother cell cytoplasm for the degradation of an as yet unidentified repressor (R) or for the activation of a activator (A) of transcription of a class of σ^K-dependent genes that includes *cdeM* and *cotA*. The encoded proteins are produced and assembled when repression is relieved or activation is triggered. Full-length YabG is represented, as it unknown whether domain A is removed. **B**: YabG is involved in processing of CspBA and SleC^FL to produce CspB and Pro-SleC. and possibly also of CotE, to separate the N-terminal chitinase (CotE^N) and the C-terminal peroxiredoxin (CotE^C) domains. YabG is also required for the degradation of SpoIVA, and possibly of CdeC. It is not known where are these proteins processed. Both the role of YabG in enforcing the proper structure of the spore and in germination are thought to contribute to host colonization. Finally, YabG is required for the transcription of at least *cdeM* and *cotA* (red arrow).

production and activation of σ^K and importantly, also required together with σ^K to enhance expression of *cdeC* [35,89]. One possibility is that SpoIIID represses and thereby delays the expression of a late class of σ^K-controlled genes, which includes *cdeM* and *cotA*, and that YabG is required for its removal. The timing of this event would be determined by the rise in *yabG* expression when the forespore turns phase-bright (Fig 8A; see above), which is also when

expression of at least *cdeM* is detected. In any event, YabG acts at the level of the *cotA* and *cdeM* promoters because expression of these genes from the *cotE* promoter bypasses the need for *yabG* for both their transcription and the association of CotA and CdeM with the spore surface (Fig 4). We cannot discard the possibility, however, that YabG is involved in the proteolytic activation of a putative activator (Fig 8A). The effect of YabG on *cdeC* expression could be indirect, in that the absence of YabG transcription of some late $\sigma^K$ dependent genes (*cdeM* and *cotA*) is blocked while the expression of earlier $\sigma^K$ targets (such as *cdeC*) is increased [25].

The control of the mother cell line of gene expression by YabG has not been reported for *B. subtilis*. In this organism, the mother cell line of gene expression is divided into several temporal and epistatic classes through the action of ancillary transcription factors that work together with $\sigma^E$ (SpoIIID, GerR) and $\sigma^K$ (GerE) [32,33,90–92]. Only SpoIIID is found in *C. difficile*. To what extent YabG influences gene expression in the mother cell and whether this role of YabG is unique to *C. difficile* remains to be studied.

### Role of YabG in host colonization

At least in strain 630Δ*erm*, spores of the *yabG* mutants show slow and less efficient germination as assessed by the drop in $OD_{600}$ of a spore suspension (S8 Fig). The plating efficiency of *yabG* spores in TA-BHI plates, however, is very close to that of WT spores, in agreement with earlier work [42]. It is therefore unclear whether inefficient germination contributes to the impaired colonization by the *yabG* mutant in the hamster assay. However, because of their increased permeability (S10 Fig), host-produced lysozyme may contribute to the germination of *yabG* spores in vivo. Full-length CotE accumulates in *yabG* spores (Fig 2A) but it is not known whether interdomain processing is a requirement for colonization and virulence [23]. The altered surface of *cdeM* spores likely explains the impaired colonization of mice [26] and the structural alterations of *yabG* spores may likewise affect the colonization ability of the mutant (Fig 8B)

The role of the YabG protease in coat/exosporium assembly, spore germination and host colonization raises the possibility that inhibitors of the enzyme may serve as chemotherapeutic agents to prevent proper morphogenesis of the *C. difficile* spore in vivo and transmission of the organism.

## Material and methods

### Ethics statement

All animal procedures were performed with prior approval from the Texas A&M Institutional Animal Care and Use Committee under the approved Animal Use Protocol number 2017–0102. Animals showing signs of disease were euthanized by $CO_2$ asphyxia followed by thoracotomy as a secondary means of death, in accordance with Panel on Euthanasia of the American Veterinary Medical Association. Texas A&M University's approval of Animal Use Protocols is based upon the United States Government's Principles for the Utilization and Care of Vertebrate Animals Used in Testing, Research and Training and complies with all applicable portions of the Animal Welfare Act, the Public Health Service Policy for the Humane Care and Use of Laboratory Animals, and all other federal, state, and local laws which impact the care and use of animals.

### Strains and growth conditions

Bacterial strains and their relevant properties are listed in S1 Table. The *Escherichia coli* strain DH5α (Invitrogen) was used for molecular cloning and BL21(DE3) (Novagen) was used for

the over-production of WT His$_{10}$-YabG and its variants and CspBA-Strep-tag by auto-induction [73]; HB101 (RP4) was used as the donor in *C. difficile* conjugation experiments [93]. Luria-Bertani medium was routinely used for growth and maintenance of *E. coli* with. ampicillin (100 μg/ml) or chloramphenicol (15 μg/ml) added if required. The *C. difficile* strains used in this study are congenic derivatives of the wild-type strain 630Δ*erm* [94]. *C. difficile* strains were grown anaerobically (5% H$_2$, 15% CO$_2$, 80% N$_2$) at 37˚C in brain heart infusion (BHI) medium (Difco) and in 70:30 plates for sporulation induction. When necessary, taurocholate (TA) (0.1% wt/vol), thiamphenicol (15 μg/ml) and/or cefoxitin (25 μg/ml) were added to the medium. For animal experiments, spores and vegetative cells were enumerated on taurocholate-cycloserine-cefoxitin-fructose agar (TCCFA) medium (Difco). All plasmids and oligonucleotide primers used are listed in S2 and S3 Tables, respectively. The inserts in all of the plasmids herein constructed were verified by DNA sequencing.

## In-frame deletion of *yabG* using CRISPR-Cas9

A CRISPR-Cas9 system [95,96] was used to create an in-frame deletion of *yabG* (*CD630_35690*) in strain 630Δ*erm* Δ*pyrE*, (S7A and S7B Fig). The forward primer to amplify the sgRNA, P5, used to direct the Cas9 nuclease to the *yabG* gene, was designed by adding the 20 nucleotides crRNA "SEED" region, identified with the Benchling CRISPR guide design tool, for the sgRNA [95,97]. P5 was used with P6 to amplify the sgRNA (159 bp). The homology arms (HA) coding for the *yabG* mutant allele were joined by PCR. The left homology arm (LHA) fragment (645 bp) was generated with P7 and P8 while the right homology arm (RHA) fragment (729 bp) was generated with P9 and P10 using as template chromosomal DNA of *C. difficile* 630Δ*erm*; the two fragments were joined by overlapping PCR. The resulting fragment (1390 bp) was cleaved with AscI and AsisI while the sgRNA fragment was cleaved using SalI and AsiSI. Both fragments were cloned between the SalI and AscI sites of pMTL431521 [95] to yield pEM28. pEM28 was introduced into 630Δ*erm* Δ*pyrE* by conjugation [98]. Transconjugants were selected as described [98]. Chromosomal DNA of the transconjugants was isolated as previously described [99]. PCR using primers P1 and P2 confirmed deletion of *yabG* in 630Δ*erm* Δ*pyrE* (S7A and S7B Fig). The resulting strain is AHCD1150 (630Δ*erm* Δ*yabG* Δ*pyrE*).

## *pyrE* reversion and *in trans*-complementation

To restore the *pyrE* gene, pMTL-YN1 [94] was conjugated into strain 630Δ*erm* Δ*pyrE* Δ*yabG* (AHCD1150). Transconjugants were plated onto minimal medium and analysed by PCR, using primers P3 and P4 (S7C and S7D Fig), to verify *pyrE* reversion. Finally, plasmid loss was tested by patch plating positive clones onto BHI supplemented with cefoxitin and thiamphenicol; this screen yielded strain AHCD1203 (630Δ*erm* Δ*yabG* *pyrE*$^+$). For complementation of the Δ*yabG* Δ*pyrE* strain in single copy at *pyrE*, *yabG* with its promoter region (a 1585 bp fragment) was PCR-amplified with P11 and P12 and the fragment inserted between the XhoI and BamHI sites of pMTLYN1C [94] to produce pEM39. In addition, *yabG*$^{C207A}$ was PCR-amplified with P11 and P13 and chromosomal DNA from 630Δ*erm* to obtain a fragment of 1192 bp. A second fragment of 419 bp was amplified using P14 and P12. The PCR fragments were fused by overlapping PCR using P11 and P12 and the resulting fragment cloned between the XhoI and BamHI sites of pMTLYN1C [94] to produce pEM41. pEM39 and pEM41 were conjugated into *C. difficile* AHCD1150 (Δ*yabG*) to introduce the *yabG* or *yabG*$^{C207A}$ alleles at *pyrE*. Both strains were analysed by PCR to verify the insertion of the allele of interest at the *pyrE* locus using primers P3 and P4 (S7C and S7D Fig). Finally, colonies of *yabG*$^C$ and Δ*yabG*$^{C207A}$ were patch plated onto BHI with cefoxitin and thiamphenicol to test for loss of the plasmids. Plasmid-cured strains were named AHCD1204 (referred to as *yabG*$^C$) and AHCD1205 (*yabG*$^{C207A}$).

### A *yabG-SNAP^Cd* transcriptional fusion

The promoter region of *yabG* was PCR-amplified from genomic DNA of 630Δ*erm* using P15 and P16, to produce a 268 bp fragment. The *SNAP^Cd* gene [29,30] was PCR amplified from pFT47 [29] using P17 and P18. The two pieces were joined by PCR and the resulting 803 bp fragment cloned between the EcoRI and XhoI sites of pMTL84121 [100] to produce pEM7 which was conjugated into the 630Δ*erm*, *sigE::ermB* (AHCD533) and *sigK::ermB* (AHCD535) strains [29] (S1 Table).

### Placing *cdeM* and *cotA* under the control of the *cotE* promoter

$P_{cotE}$-*cotA* was constructed by PCR-amplifying the *cotE* promoter from the genomic DNA of *C. difficile* 630 using primers P49/P50 and P51/P52. The two fragments were joined by PCR to produce a product of 1246 bp that was inserted between the EcoRI and HindIII sites of pMTL84121 to yield pSR77 which was conjugated into *C. difficile* 630Δ*erm*, Δ*yabG* and Δ*yabG pyrE::yabG^C207A* strains yielding AHCD1502, AHCD1503, and AHCD1504, respectively. $P_{cotE}$-*cdeM* was constructed by PCR-amplifying the *cotE* promoter from the genomic DNA of *C. difficile* 630 using primers P53/P54 and P54/P55. The two pieces were joined by PCR and inserted between the EcoRI and HindIII sites of pMTL84121 to give pCAF3. Plasmid pCAF3 was conjugated into 630Δ*erm* (WT) producing strain AHCD 817.

### Translational YabG-SNAP^Cd fusions

To construct a C-terminal SNAP^Cd-tag fusion to *yabG^WT*, the sequence containing the *yabG* promoter and coding regions without the stop codon was PCR-amplified using P15 and P30 and chromosomal DNA from 630Δ*erm* [29] and cloned into pFT58 [29] yielding pEM5 ($P_{yabG}yabG^{WT}$-*snap^Cd*). To construct a C-terminal SNAP^Cd-tag fusion to *yabG^C207A*, the *yabG* promoter region was PCR-amplified using P15 and P13. The *yabG^C207A* coding sequence was amplified using primers P30 and P14 The two fragments were joined by PCR and the resulting fragment was cleaved using EcoRI and BamHI and cloned into pFT58. This produced plasmid pEM40 ($P_{yabG}$-*yabG*^C207A- *snap^Cd*).

### *cspBA-strep-tag* into pACYCDuet-1

The *cspBA* coding sequence was PCR-amplified using primers P31 and P32. The resulting fragment was inserted between the NcoI and NotI sites of pACYC-duet (Novagen) to create the *cspBA-strep-tag*-expression plasmid pEM23.

### *his₁₀-yabG* in pET16b

The *yabG* coding sequence was PCR-amplified using primers P33 and P34. The resulting fragment was inserted between the BamHI and XhoI sites of pET16b (Novagen) to create pEM6. Mutations generating single Ala substitutions were introduced in *yabG* using primer pairs P14/P13, P37/P38, P39/P40, P41/P42, P43/P44 and pEM6 as the template; this created plasmids pEM12, pEM13, pEM21, pEM38, and pEM24, respectively.

### Spore production and purification

Cultures (150 ml) in BHI medium were incubated at 37°C under anaerobic conditions for 7 days. Cells were collected by centrifugation (at 4800*xg*, for 10 min, 4°C) resuspended in cold water and stored for 48 hours at 4°C. The suspensions were then collected by centrifugation and the sediment resuspended in 1 ml of PBS-tween 20. Spores were purified on density

gradients of Gastrografin (Bayer) [101]. The spore titer in the suspension was measured spectrophotometrically at an $OD_{580}$.

## Spore fractionation and mass spectrometry

Spores were resuspended in 50 μl of decoating buffer (10% glycerol,4%SDS,10% β-mercaptoethanol, 1mM DTT, 250 Mm Tris Ph 6,8) to a final $OD_{580}$ of 4.0. The suspension was boiled for 5 minutes and the spores collected by centrifugation. The supernatant, corresponds to a cortex/coat/exosporium fraction. The spore sediment was washed twice with PBS with 0.1% Tween-20, and incubated with 50mM Tris-HCl pH 8 with 2mg/ml lysozyme for 2 h at 37°C to digest the spore cortex peptidoglycan and release core/cortex-associated proteins.

## Whole cell extracts and immunobloting

Whole cell extracts were prepared from *E. coli* or sporulating cells of *C. difficile* as described before [29]. The whole cell extracts and the cortex/coat/exosporium and core/cortex fractions of spores were analysed by 15% SDS-PAGE and immunoblotting. Antibodies were used at the following dilutions: anti-YabG (1:1000), anti-CdeC (1:500), anti-CdeM (1:15000), anti-CotA (1:1000), anti-CspB, anti-CspC, and anti-SleC (1:3000), anti-GPR (1:10000), anti-SNAP-tag at 1:1000 and anti-His tag (at 1:1000). A rabbit secondary antibody conjugated to horseradish peroxidase (Sigma) was used at a dilution of 1:5000; an anti-mouse IgG (whole molecule)-peroxidase (Sigma) was used at a dilution of 1:2000 for the detection of the SNAP-tag. All the immunoblots were developed with the Super Signal Pico Plus Chemiluminescent Substrate (Thermo Scientific). For protein identification, protein bands were excised from Coomassie-stained gels, digested with trypsin and analysed by matrix-assisted laser desorption ionization (MALDI) mass spectrometry.

## RNA extraction and quantitative RT-PCR analysis

Sporulating cells were collected from cultures of the 630Δ*erm* strain and the Δ*yabG* mutant after 14 h and 20 h of growth on 70:30 agar plates. RNA was extracted from three independent cultures, using the RNeasy mini kit (Qiagen), according to the manufacturer's instructions. cDNAs synthesis and real-time quantitative PCR were as previously described [102,103]. In each sample, the quantity of cDNAs for each gene was normalized to the quantity of cDNAs of the *rpoC* gene. The primers used for each marker are listed in S3 Table. The relative transcript changes were calculated using the $2^{-\Delta\Delta Ct}$ method as described.

## Spore lysozyme resistance and lysozyme permeability assays

Lysozyme resistance was assessed based on the method described in [79]. Spores ($1x10^7$) of the WT strain 630Δ*erm*, *yabG* mutants or the complementation strain were resuspended in a volume of 0.5 ml PBS buffer and incubated for 30 min at 37°C with lysozyme (250 μg/ml) followed by heat treatment for 10 min at 80°C, serial dilution and plating on BHI agar plates containing 0.1% TA to determine surviving colony forming units (CFU)/ml. The number of phase-dark spores resulting from exposure to lysozyme (under the conditions above, but not heat treated) was also scored using phase contrast microscopy.

## Spore Germination and DPA content

Density-gradient-purified spores were resuspended in BHI to a final $OD_{600}$ of 1 and heat activated for 10 min at 80°C. Taurocholic acid (TA) (Sigma-Aldrich) in BHI was added to a final concentration of 0.5%. Germination was followed under anaerobic conditions (5% $H_2$, 15%

$CO_2$, 80% $N_2$) at 37˚C, by following the decrease in the $OD_{600}$ of the spore, until no significant changes were detected (data is shown for the first 90 min). The DPA released was measured by incubating spores at 37˚C in PBS supplemented with 0.5% TA for 1 hour as described [51]; the same amount of spores was incubated at 37˚C in PBS without TA addition (negative control) or at 100˚C with 0.5% TA (positive control). The spore suspension was centrifuged (at 6000 x$g$ for 3 min at room temperature) and the released DPA estimated by measuring the $OD_{270}$ of the supernatant. The efficiency of spore germination was also estimated using a plate assay as described [42].

## Microscopy

For SNAP labelling, samples were withdrawn from sporulating cultures at 14h and 20h growth into 70:30 media [53], incubated with the TMR-Star substrate (New England Biolabs), processed for phase contrast and fluorescence microscopy and imaged as described [29]. The intensity of the FM4-64 signal in the spore polar appendage was quantified using *ImageJ* (http://rsbweb.nih.gov/ij/) by drawing a 0.25 μm diameter circle in the appendage region. The data from three independent experiments was represented using SuperPlots [104]. For the characterization of spores of strains producing either YabG[WT]- or YabG[C207A]-SNAP[Cd] in the WT or Δ*yabG* background, with TMR-Star as described above and with the membrane dye Mitotracker Green (MTG, 0.5 ng/mL; Invitrogen). Super-resolution Structured Illumination Microscopy (SR-SIM) as conducted as described previously [99].

## Scanning (EM) and transmission electron microscopy (TEM)

Thin sectioning TEM of density gradient-purified *C. difficile* spores was as described previously [29]. For SEM, highly purified spores were fixed with 2.5% glutaraldehyde, 1% formaldehyde, 0.1M phosphate buffer during 30 min at room temperature. Samples were washed 3 times with 0.2 M $CaCl_2$ and 0.1% Tween 20, dehydrated at room temperature with increasing concentrations of ethanol washes up 100% and finally dehydrated in 100% acetone during 5 min. Samples were mounted on glass slides and covered with gold. Scanning electron microscopy was performed on a Hitachi SU-8010 field emission gun operated at 1.5kV.

## AlphaFold2 modelling

AlphaFold2 [60] was used to predict the structure of *C. difficile* YabG. No template structures were used in the prediction, which was iterated for up to 48 recycles, followed by energy refinement with AMBER using default settings implemented in ColabFold [105] and using MMseqs2 for creating multiple sequence alignments [106]. The confidence of the modelling was assessed by the pLDDT metric and the predicted alignment error (PAE), *i.e.*, the uncertainty about the interface (S2 Fig). Values of pLDDT > 90 are expected to have high accuracy. Five YabG models were generated; rank 1 model is represented in Fig 1C. The statistics associated with each of the five models are shown in S2 Fig.

## Animal studies

All animal studies were performed with prior approval from the Texas A&M University Institutional Animal Care and Use Committee. Female Syrian golden hamsters, 80g - 120g, were housed individually in cages and had *ad libitum* access to food and water for the duration of the experiment. To induce susceptibility to *C. difficile* infection, hamsters were gavaged with 30 mg/kg clindamycin [84,85]. After 5 days, hamsters were gavaged with 1.000 spores of WT *C. difficile* (n = 7) or *C. difficile* Δ*yabG* (n = 8) and monitored for signs of disease (lethargy,

poor fur coat and wet tail). Hamsters showing signs of disease were euthanized by $CO_2$ asphyxia followed by thoracotomy as a secondary means of death in accordance with Panel on Euthanasia of the American Veterinary Medical Association. Fecal samples were collected daily and weighed, suspended in 1 mL of sterile water and dissociated with a pipette. The samples were serially diluted and plated on to TCCFA agar medium to enumerate both vegetative *C. difficile* cells and spores. CFUs were tabulated and expressed as CFU/g feces. The data has been set to the median values and the limit of detection was $10^3$ CFU/g.

## Supporting information

**S1 Fig. *B. subtilis* and *C. difficile* YabG.** Schematic representation of the *yabG* region of the *B. subtilis* (**A**, at 1.23 map units; close and to the right to the origin of chromosome replication, *ori*) and *C. difficile* chromosomes (**B**, at 97.19 map units, close but to the left of *ori*). The bottom panel represents the predicted structural organization of the two proteins (see also Fig 1C).
(TIFF)

**S2 Fig. Statistics for the AlphaFold-generated structural model of *C. difficile* YabG. A**: Sequence coverage; number of sequences and sequence identity to the query. **B**: Local Distance Difference Test (IDDT) per position. **C**: rank of the five models generated. The plots shown are reproduced from the output of the ColabFold server [105].
(TIFF)

**S3 Fig. Domain A of YabG shares structural similarity with SH3 domains.** The Alpha-Fold2-generated model of the A domain of YabG (orange) is superimposed onto the NMR solution structure of the SH3 protein PetP a subunit of the cyanobacterial cytochrome *b6f* (**A**; pdb code 2n5u) and the crystal structure of *E. coli* HspQ (**B**; pdb code 5ycq; also known as YccV). Panel **A** shows the ensemble of the NMR structures determined for the PetP protein.
(TIFF)

**S4 Fig. The B domain of *C. difficile* YabG.** The AlphaFold2-generated model of YabG (green) is superimposed onto the crystal structures of CheY (pdb identifier: 5chy; brown) and KdpE of *E. coli* (pdb identifier: 4I85; black). The panels on the right show a trace of the structures to reveal the position of the His161, Asp162 and Cys207residues in YabG.
(TIFF)

**S5 Fig. YabG$^{C119A}$ shows auto-proteolytic activity. A**: His$_{10}$-YabG$^{WT}$ (WT), His$_{10}$-YabG$^{C119A}$ (C119A) and His$_{10}$-YabG$^{C207A}$ (C207A) were produced in *E. coli*. The proteins in whole cell extracts were resolved by SDS-PAGE and the gels stained with Coomassie or subject to immunoblotting with an anti-His$_6$ antibody. **B**: The alignments show blocks of amino acids conserved between the YabG proteins of *B. subtilis* (*B. s.*) and *C. difficile* (*C. d.*) in the vicinity of the processing sites determined for the *B. subtilis* protein. Numbering is from the first residue of the proteins.
(TIFF)

**S6 Fig. *yabG* is under the control of σ$^E$ and σ$^K$. A**: The panel shows the *yabG* regulatory region and highlights the putative -10 and -35 promoter elements with the consensus for σ$^{E/K}$ binding indicated below (bases that match the consensus are in red; [76]). The ribosome binding site (RBS), the *yabG* start codon and the stop codon of the *CD630_35700* gene are also indicated. **B**: Expression of a P$_{yabG}$-SNAP$^{Cd}$ transcriptional fusion in sporulating cells of the WT (630Δerm) and in congenic *sigE* and *sigK* mutants. The cells were collected after 14h or 20h of growth in 70:30 agar plates [53], stained with the TMR-Star SNAP substrate and

examined by phase contrast and fluorescence microscopy. The blue arrowheads indicate the position of the forespore in the phase contrast and in the autofluorescence images (green channel); the yellow arrows show the mother cell-specific SNAP$^{Cd}$-TMR-Star signal (red). Note the disporic sporangium in the *sigE* mutant. The numbers indicate the percentage of sporangia at the indicated stages showing P$_{yabG}$-SNAP$^{Cd}$ expression. A representative stage for each mutant was selected. At least 50 cells were scored for each strain in each of three independent experiments. Scale bar, 1μm. **C**: Intensity of the fluorescence signal per cell for the P$_{yabG}$-SNAP$^{Cd}$ fusion in the mother cell during asymmetric division and engulfment (AD/Eng), and in sporangia of phase-dark, phase-grey or phase-bright forespores in the WT (top) or in sporangia of phase-grey and phase-bright forespores in a *sigK::ermB* mutant (bottom). No signal was detected in *sigE::ermB* sporangia. Fluorescence intensity is shown in arbitrary units (AU). SuperPlots were used to represent the data from three biological replicates; each dot corresponds to one cell, color-coded by experiment. The large circles represent the means from each experiment which were used to calculate the mean and standard error of the mean (horizontal lines) for the ensemble of the three experiments. Statistical analysis was carried using a Student's t-test (right panels) or an one-way ANOVA followed by Tukey's multiple comparations test (left panels). *, $p < 0.05$; ***, $p < 0.0001$.
(TIFF)

**S7 Fig. In-frame deletion of the *yabG* gene in *C. difficile* 630Δ*erm* using CRISPR-Cas9 genome editing. A**: Genetic organization of the *C. difficile* chromosome in the vicinity of *yabG*. Plasmid pEM28 codes for the single guide RNA carrying the seed region of 20 nucleotides that directs Cas9 to *yabG* [95,108]. The position of the sequences recognized by primers P1 and P2 is also shown. **B**: Chromosomal DNA was prepared from the WT and a thiamphenicol resistant *C. difficile* conjugant and screened by PCR. The presence of the PCR product of 1515 bp, as opposed to 2184 bp for the WT, identifies the *yabG* in-frame deletion mutation. **C**: Schematic representation of Δ*pyrE* reversion using homologous recombination between pMTL-YN1 and the genome of 630Δerm Δ*yabG*Δ*pyrE*. **D**: Δ*yabG pyrE+* isolates was screened by PCR for the presence/absence of a reverted *pyrE* gene. The Δ*yabG pyrE+* strain results from recombination of pMTL-YN1 at the *pyrE* locus (**C**). In trans complementation of the Δ*yabG* in-frame deletion was accomplished by introducing the WT copy of *yabG* (*yabG*$^C$) at the *pyrE* locus using allelic exchange according to the scheme in C using pEM39; similarly, but using pEM41, the *yabG*$^{C207A}$ allele was transferred to the *pyrE* locus (see also the Material and Methods section).
(TIFF)

**S8 Fig. Spore germination is impaired in Δ*yabG* and *yabG*$^{C207A}$ mutants. A**: Purified spores were heat activated and BHI-TA (0.5%) was added (open symbols). Germination was followed by the decrease in the optical density of the spore suspension at 600 nm and expressed as the percentage of the initial OD$_{600}$. In control experiments, the spores were maintained in BHI, with no TA (closed symbols). **B**: Phase contrast of purified spores incubated in BHI in the presence of 0.5% TA. The numbers refer to the percentage of phase-dark spores. Scale bar, 1 μm. **C**: Top panel: germination efficiency for spores of the indicated strains, expressed as the fraction of the WT; middle panel: the DPA released after TA-induced germination at 37°C is shown for spores of the indicated strains and it is expressed as the ratio between the OD$_{270}$ and the initial OD$_{600}$ of the spore suspension; bottom panel: spores resuspended in PBS supplemented with 0.5% TA were boiled to determine the total DPA content (total DPA released; blue bars), expressed as a percentage of DPA content of the WT; The results are the average of three independent experiments; statistical significance was determined using ANOVA and Tukey's test. **D:** Immunoblotting of proteins extracted from density-gradient purified spores

produced by the WT, *ΔyabG*, *yabG^C207A* and the complementation strain (*yabG^C*). The proteins were resolved by SDS-PAGE and the gels subject to immunoblot analysis with anti-CspB, anti-SleC and anti-CspC antibodies. The experiments were repeated at least three times.
(TIFF)

**S9 Fig. Phase contrast microscopy of spores during germination in a rich medium supplemented with taurocholic acid.** Δ*yabG*, yabG^C207A, yabG^C, and WT highly purified spores were heat activated for 10 min at 80˚C and TA added (to 0.5%) in rich media under anaerobic conditions. Samples were collected at the indicated times (in min) after TA addition and imaged by phase contrast microscopy. Red arrowheads indicate phase-bright free spores while yellow arrowheads show phase-dark, germinating spores. Numbers refer to the percentage of phase-dark spores scored at the indicated times: The data refers to one of three independent experiments. Scale bar 1 μm.
(TIFF)

**S10 Fig. *yabG* spores are more permeable to lysozyme. A:** Purified spores of the indicated strains were incubated for 30 min at 37˚C in the absence or in the presence of lysozyme (250 μg/ml). Following incubation, the samples were examined by phase contrast microscopy (left panel) and the percentage of phase-dark spores scored (right panel). The yellow arrowheads point to phase dark-spores. Scale bar, 1 μm. **B**: Shows the percentage of phase-dark spores before and after treatment with lysozyme 250 μg/ml. The results are the average for three independent experiments; 170–250 spores were counted per each strain in each experiment. The p-value is indicated for all comparisons whose differences were found to be statistically significant using ANOVA and Tukey's test (*, $p \leq 0.05$).
(TIFF)

**S11 Fig. YabG has a role in the formation of a spore polar appendage. A:** Density gradient purified spores of the indicated strains (WT, *yabG^C*, Δ*yabG* and *yabG^C207A*) were examined by phase contrast and fluorescence microscopy. The figure shows three panels, aligned vertically, for spores of each strain. Yellow arrowheads, polar appendages in the WT of *yabG^C* strain; red arrowheads, the squarish appendage in the two *yabG* mutants. Scale bar, 1 μm. **B**: Percentage of spores with appendages for each of the indicated strains. Spores were considered to possess an appendage if its length was $\geq 0.25$ μm. At least 60–80 spores were scored for each strain; three biological replicates were performed. Statistical significance was determined using ANOVA and Tukey's test (* $p < 0.05$). **C**: Spores of the indicated strains were stained with FM4-64 and imaged by phase contrast and fluorescence microscopy. The arrowheads (yellow in the fluorescence images and black in the phase contrast images, point to the appendage region). Scale bar, 1 μm. **D:** Quantification of the FM4-64 intensity signal associated with the appendage region of purified spores of WT, Δ*yabG*, *yabG^C* and *yabG^C207A* strains. Note that the staining of the appendage with FM4-64 is higher for spores of the Δ*yabG* and *yabG^C207A* mutants compared to the WT and *yabG^C* spores. For each strains, 30–50 spores were scored from three independent experiments. SuperPlots were used to represent the data; each dot corresponds to one cell, color-coded by experiment. The large circles represent the means from each experiment which were used to calculate the mean and standard error of the mean (horizontal lines) for the ensemble of the three experiments. Statistical analysis was carried out using using ANOVA and Tukey's test. *, $p < 0.05$; **, $p < 0.001$.
(TIFF)

**S12 Fig. Transmission electron microscopy of *yabG* spores.** Purified spores of the WT (630Δ*erm*), Δ*yabG* mutant and the complementation strain (*yabG^C*) were analysed by thin sectioning TEM. Red arrowheads point to the coat region, blue arrowheads to the electron dense

exosporium and yellow arrowheads to the region of transition between the coat and the appendage. The regions delimited by the red and yellow circles in the top panels are magnified on the bottom panels, as indicated. Cr, spore core; Cx, cortex; Ap, appendage region. The numbers refer to the percentage of spores in which the coat is detached from the cortex (red) or which show a prominent polar appendage, with a lamellar structure (blue) (see also Fig 2C). Between 60–95 spores were scored for each strain in two independent experiments. Scale bar, 500 nm (top panels) and 100 nm (all other panels).
(TIFF)

**S13 Fig. Localization of YabG-SNAP$^{Cd}$. A.** Localization of YabG$^{WT}$-SNAP$^{Cd}$ and YabG$^{C207A}$-SNAP$^{Cd}$ in the WT and in the Δ*yabG* mutant. Cells were collected after 14h of growth in 70:30 agar plates, stained with the SNAP substrate TMR-Star and examined by phase contrast and fluorescence microscopy (red channel for the TMR signal and the green channel for autofluorescence signal). The numbers refer to the percentage of cells at the represented stage exhibiting SNAP fluorescence. The data shown are from one experiment of three independent experiments. For each strain, at least 145 cells were scored per time point. Scale bar, 1 μm. **B:** Accumulation of YabG-SNAP$^{Cd}$ and YabG$^{C207A}$-SNAP$^{Cd}$ in sporulating cells at 14h in the WT strain 630Δ*erm*, and in the Δ*yabG* mutant. Whole cell extracts were prepared, the proteins resolved by SDS-PAGE and the gels subject to immunoblotting with anti-SNAP antibodies. The red arrowhead point to the position of YabG$^{C207A}$-SNAP (52 kDa) and asterisks indicate possible degradation products that include the SNAP moiety (~19.4 kDa). Asterisks show the position of degradation products or cross-reactive species.
(TIFF)

**S14 Fig. YabG affects expression of its coding gene. A:** Samples from cultures of the WT, and congenic Δ*yabG* and *yabG$^{C207A}$* mutants expressing a P$_{yabG}$-*SNAP$^{Cd}$* transcriptional fusion were collected after 20h of growth in 70:30 agar plates and labelled with TMR-Star and imaged by phase contrast and fluorescence microscopy. The numbers in the panels show the percentage of sporangia with signal from the SNAP$^{Cd}$-TMR complex. Expression of the P$_{yabG}$-*SNAP$^{Cd}$* fusion was scored in early stage sporangia (most with phase-dark spores), and in sporangia of phase-grey and phase-bright spores. White arrowheads, mother cell in the phase contrast images; yellow arrowheads, signal from SNAP$^{Cd}$-TMR. Scale bars, 1 μm. **B:** Shows the distribution of the fluorescence signal (in arbitrary units, AU) in the different types of sporangia considered. For each stage, 50 sporangia were scored, per strain in each of three biological replicates. SuperPlots were used to represent the data, with each dot corresponding to one cell and the three experiments shown with different colors. The large circles represent the means from each experiment and were used to calculate the mean and standard error of the mean (horizontal lines) for the collective of the three experiments. Statistical analysis was carried out using a Student's t-test. * indicates p<0.05. Scale bar, 1 μm.
(TIFF)

**S15 Fig. Localization of YabG$^{WT}$- or YabG$^{C207A}$-SNAP$^{Cd}$ in mature spores.** Localization of YabG$^{WT}$- or YabG$^{C207A}$-SNAP$^{Cd}$ in mature spores using super resolution structured illumination microscopy (SR-SIM), in either the WT or Δ*yabG* backgrounds. The spores were stained with the membrane dye MTG (green) and with TMR-Star (red) prior to imaging. The blue arrowheads point to the appendage-associated signal; the white arrowheads point to the signal in other regions of the spore (see also Fig 6A). Scale bar, 500 nm.
(TIFF)

**S1 Table. Bacterial strains.**
(PDF)

**S2 Table. Plasmids used in this study.**
(PDF)

**S3 Table. Oligonucleotides used in this study.**
(PDF)

## Acknowledgments

We thank Erin Tranfield and Ana Sousa of the Electron Microscopy Facility at the *Instituto Gulbenkian de Ciência* (www.igc.gulbenkian.pt) for technical expertise and sample processing and Aimee Shen and Cecile Morlot for helpful discussions. We thank Fernando Cruz for support on the statistical analysis and Ana Henriques for help with art work.

## Author Contributions

**Conceptualization:** Adriano O. Henriques.

**Formal analysis:** Eleonora Marini, Carmen Olivença, Manuel N. Melo, Joseph A. Sorg, Adriano O. Henriques.

**Funding acquisition:** Joseph A. Sorg, Mónica Serrano, Adriano O. Henriques.

**Investigation:** Eleonora Marini, Carmen Olivença, Sara Ramalhete, Andrea Martinez Aguirre, Patrick Ingle, Manuel N. Melo, Wilson Antunes, Guillem Hernandez.

**Project administration:** Adriano O. Henriques.

**Resources:** Adriano O. Henriques.

**Supervision:** Nigel P. Minton, Tiago N. Cordeiro, Joseph A. Sorg, Mónica Serrano, Adriano O. Henriques.

**Writing – original draft:** Adriano O. Henriques.

**Writing – review & editing:** Joseph A. Sorg, Mónica Serrano, Adriano O. Henriques.

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
