## [Decision Letter · Decision Letter 0]

11 May 2023

Dear Prof Henriques,

Thank you very much for submitting your manuscript "A sporulation signature protease is required for assembly of the spore surface layers, germination and host colonization in Clostridioides difficile" for consideration at PLOS Pathogens. As with all papers reviewed by the journal, your manuscript was reviewed by members of the editorial board and by independent reviewers. In light of the reviews (below this email), we would like to invite the resubmission of a significantly-revised version that takes into account the reviewers' comments.

The manuscript has been reviewed by two experts on Clostridioides difficile biology. While those Reviewers are generally positive, Reviewer 2 has a few scientific comments that would likely require some additional experimentation if the results are not already available. This Reviewer also raises some questions about the alpha fold modeling that should be considered. In addition to those Science comments, both Reviewers raised concerns about the manuscript itself. These concerns are specified in their reviews. But, in particular, note that Reviewer 2 believes the manuscript would be strengthened by condensing the text. Reviewer 1 believes that the Materials and Methods for supplemental experiments should be moved into the main text. This comment can be extended to include moving the "Results and Discussion" section in Supplemental Materials to the main text to conform with the normal formatting for PLoS Pathogens articles. Please check the journal homepage to see the normal journal style. Generally Supplemental Materials should be restricted to Tables and Figures, with legends.

We cannot make any decision about publication until we have seen the revised manuscript and your response to the reviewers' comments. Your revised manuscript is also likely to be sent to reviewers for further evaluation.

Sincerely,

Bruce A. McClane

Academic Editor

PLOS Pathogens

Michael Wessels

Section Editor

PLOS Pathogens

Kasturi Haldar

Editor-in-Chief

PLOS Pathogens

orcid.org/0000-0001-5065-158X

Michael Malim

Editor-in-Chief

PLOS Pathogens

orcid.org/0000-0002-7699-2064

The manuscript has been reviewed by two experts on Clostridioides difficile biology. While those Reviewers are generally positive, Reviewer 2 has a few scientific comments that would likely require some additional experimentation if the results are not already available. This Reviewer also raises some questions about the alpha fold modeling that should be considered. In addition to those Science comments, both Reviewers raised concerns about the manuscript itself. These concerns are specified in their reviews. But, in particular, note that Reviewer 2 believes the manuscript would be strengthened by condensing the text. Reviewer 1 believes that the Materials and Methods for supplemental experiments should be moved into the main text. This comment can be extended to include moving the "Results and Discussion" section in Supplemental Materials to the main text to conform with the normal formatting for PLoS Pathogens articles. Please check the journal homepage to see the normal journal style. Generally Supplemental Matierials should be restricted to Tables and Figures, with legends.

Reviewer's Responses to Questions

**Part I - Summary**

Reviewer #1: In this study, Marini et al. investigate the role of YabG in C. difficile spore morphogenesis and virulence. They identify several YabG substrates, including YabG itself, CspBA, pre-pro-SleC, and CotE. The latter three are absent from B. subtilis, which shows that C. difficile and B. subtilis YabG have distinct substrates. In a hamster model of CDI, a yabG mutant shows wild-type levels of virulence but much lower levels of colonization. Overall, the experiments are well designed, results and interpretations are clear, and the manuscript is generally well written.

Weaknesses are minor. I think the importance and novelty of the study lies in what distinguishes YabG from C. difficile from that of B. subtilis, the authors might emphasize these differences, such as in the abstract.

Reviewer #2: The manuscript by Marini et al. defines the role of a conserved sporulation-specific protease, YabG, during the morphogenesis and germination of Clostridioides difficile spores. The paper establishes that loss of YabG or its catalytic activity lead to defects in coat and exosporium assembly as well as alternations in spore germination (the effects of YabG on the latter process have been described in detail by Shrestha and Sorg in this journal in 2019). Importantly, the authors demonstrate that the structural defects of yabG mutant spores (and potentially their altered germination properties) lead to a significant colonization defect in a hamster model of infection, highlighting the critical role that this protease plays in establishing C. difficile infection. There is little impact on virulence because hamsters are so sensitive to toxin levels that the greatly reduced colonization observed with the yabG mutant still causes death in all the hamsters.

The authors also identify a novel role for this morphogenetic protease in regulating the transcription of late sporulation genes, a finding that adds an important new layer to the regulation of C. difficile sporulation. While multiple nested transcriptional loops that control the timing and expression levels of late-acting sporulation factors to ensure the fidelity of the spore assembly process have been established in B. subtilis, there are only two known nested transcriptional loops in C. difficile: SpoIIID in the mother cell and SpoVT in the forespore. The authors’ findings indicate that C. difficile has linked the activity of a morphogenetic protease to the assembly of robust spores that can withstand the harsh environment of the gut to establish colonization. This is a point that they could more explicitly state in discussing the significance of their findings. By condensing their discussion of these transcriptional analyses, they would be able to more clearly and succinctly communicate the impact of this particular finding to the reader.

Indeed, while this work is thorough, carefully conducted, and clearly explained

the manuscript would benefit from being condensed. There is too much speculation of the AlphaFold model, especially given that the model does not convincingly match their mutagenesis analyses (described in more detail below). Suggestions for condensing the manuscript are provided here:

- Combine Figures 1 and 2 and condense all the AlphaFold descriptions and interpretations

- Move 2A to the Supplement

- Move Figure 3 to the Supplement and condense the Discussion

- Move Figure 8 to the Supplement

- Discussion could be greatly condensed; speculation on YabG’s catalytic center should be removed

A key point that the authors should address in a revised manuscript is the clear substrate preference of YabG for cleavage after Arg residues. This has been shown with B. subtilis YabG (Yamazawa et al. 2022) as well as with C. difficile YabG’s substrates SleC and CspBA (Shrestha et al. 2019). Are there arginine residues with YabG where it is likely autoprocessing as was observed for B. subtilis YabG?

AA detail that is missing from the Materials and Methods is how many biological replicates were analyzed in their image quantifications. This information should be provided throughout the manuscript figure legends and in the Materials and methods. Importantly, the use of Superplots would greatly increase the rigor of their data analysis. If only one biological replicate was quantified, this should be indicated. Notably, no statistical analyses can be performed on one replicate even if hundreds or thousands of measurements were made for an n=1 sample. If multiple biological replicates were quantified, the statistical analyses should be performed on the three replicate values rather than the number of events quantified. https://rupress.org/jcb/article/219/6/e202001064/151717/SuperPlots-Communicating-reproducibility-and

**Part II – Major Issues: Key Experiments Required for Acceptance**

Reviewer #1: I do not believe additional experiments are needed.

Reviewer #2: My most significant concern with the manuscript is that the AlphaFold model shown in Figures 1D and S4 do not show an active site with the catalytic residues aligned in a manner that is consistent with most proteases. The catalytic residues of papain are within much closer proximity than the distance between the likely catalytic residues determined experimentally and via sequence alignment. The putative catalytic Aspartate 162 is even oriented away from the catalytic His161 and Cys207 residues. Thus, language like “Cys 207, His161, and Asp162 can be superimposed onto the catalytic triad” is not reflected in the models shown. Given that the AlphaFold model does not give an accurate prediction of the active site, the proposed similarity of YabG to the receiver domain of response regulators is less convincing, especially since it does not appear to be regulated by phosphorylation and the authors do not perform structure-guided mutagenesis of the different domains.

For the fractionation experiments, the authors indicate that their two fractions represent the coat/exosporium and cortex/cytosol. However, it is likely that their coat/exosporium fraction likely contains cortex proteins. The authors should blot for SleC to define their fractions, since C. difficile SleC is the sole protein known to be localized specifically to the cortex layer (Baloh et al. 2022).

The authors should formally report whether their fluorescent-protein tagged YabG constructs are functional.

**Part III – Minor Issues: Editorial and Data Presentation Modifications**

Reviewer #1: Can YabG be attributed a catalytic triad if one of the triad residues is dispensable for activity?

Lines 447-450, are these percentages relative to total or to the number of spores with polar appendages? Also, I suggest putting the information on adherence of the coat to the cortex together, i.e. the information in lines 450-452 and 462-463 should be in the same paragraph for easier comparison

In the animal experiments, there appear to be few animals in the group infected with the mutant. This does not appear to be due to animal survival since there are no differences in survival between the WT and mutant infected groups. Is it that there are mutant-infected animals with CFU below the limit of detection? If so, were those data included in determination of medians?

Animal experiment data – please indicate what is shown in Fig 9B. Presumably the bars represent medians. What is the limit of detection?

How many independent samples/cells were analyzed by immunoblot and microscopy? This information is needed so the reader can evaluate the representativeness and accuracy of percentages stated in the text.

Consider the content of the materials and methods in the main paper versus supplemental. For example, I think experimental parameters for microscopy are more important in the main text than strain generation.

Other minor points/recommended edits:

Please remove superfluous commas throughout, and there are multiple places where commas are needed for clarity.

L95, recommend changing to “certain bile salts”

L97, differentiate into spores

There are numerous typos and other errors in the Discussion that need correcting, including but not limited to lines 708, 710, 727, 729, 775, 811

L164, “mutations in yabG were found to render…”

L166, typo

L193, “activity that causes”

L212-216 – sentence is difficult to follow. Recommend removing all words after HspQ or breaking up the sentence into two

L240-250 – the organization of this section makes it difficult to follow. Suggest reorganizing to more explicitly delineate what is and is not similar to CheY

L266, recommend removing “using auto-induction” since expression appears to be under the control of a T7 promoter

Fig 1 – consider swapping the order of the 2nd and 3rd alignments in panel B so that they are in numerical order

Fig 4C, the arrows in the bottom panels look orange/brown, but text states they’re black

Fig 5C – statistical analysis is needed for the qRT-PCR data

Text in lines 803-806 should be placed earlier in this section

L813, reword to “This reduced germination might…”

L821, “…CotE contributes to…”

Reviewer #2: Line 87: please add a period after “therapy”

Line 94: please revise to indicate that outgrowing cells are generated from germinating spores

Line 122: Consider changing “and” to “, which”

Line 123: please delete the comma after “goal”

Line 143: please add a comma after “shown”

Line 146: change “assembly” to “assembling”

Line 155: please add “accumulate in their unprocessed forms”

Line 166. It would be helpful to add a sentence describing the proposed role of CspA in sensing co-germinant.

Line 172: consider condensing “analysis of an in-frame…” to “Loss of YabG or its catalytic activity alters the assembly of the exosporium

Line 174: YabG is important for regulating the sensitivity of C. difficile spores to “co-germinants”

Lines 175-177: The sentences starting with “These phenotypes” could be removed.

Line 197: remove the comma before “reveals”

Line 223: “pneumoniae” is misspelled

Line 286: it is unclear whether separation of CspB and CspA is required for germination because even in the absence of YabG or if YabG-dependent cleavage sites, CspBA still undergoes processing at aberrant sites.

Line 729: add a period after “YabG” and “The”

Line 1294: change “unspecific” to “non-specific”

Line 1314: “not” should be changed to “no”

Line 1325: please add a comma before “while”

Line 1354: Please change “Total RNAs was” to “Total RNA was extracted”

Please note that “phase-dark”, “phase-bright”, “phase-contrast”, “fold-change,” should be hyphenated.

PLOS authors have the option to publish the peer review history of their article (what does this mean?). If published, this will include your full peer review and any attached files.

Reviewer #1: No

Reviewer #2: No
---

## [Decision Letter · Decision Letter 1]

9 Oct 2023

Dear Prof Henriques,

We are pleased to inform you that your manuscript 'A sporulation signature protease is required for assembly of the spore surface layers, germination and host colonization in Clostridioides difficile' has been provisionally accepted for publication in PLOS Pathogens.

In addition, I recommend that you consider incorporating the additional reviewer suggestions to enhance the clarity of the manuscript.  These minor changes can be made when you are completing the editorial and formatting requests that will follow.  Please carefully proofread your final submission as it will not undergo additional editorial review before publication.

Best regards,

Michael Wessels

Section Editor

PLOS Pathogens

Kasturi Haldar

Editor-in-Chief

PLOS Pathogens

orcid.org/0000-0001-5065-158X

Michael Malim

Editor-in-Chief

PLOS Pathogens

orcid.org/0000-0002-7699-2064

Reviewer Comments (if any, and for reference):

Reviewer's Responses to Questions

**Part I - Summary**

Reviewer #2: YabG protease resubmission review

The authors have addressed the concerns/ questions raised by the Reviewers. The manuscript is very comprehensive. My only suggestion is that the results are described in some ways too thoroughly at times so the salient points of the manuscript get diluted by the extensive descriptions of the results. For example, the extensive description of the data regarding SigE and SigK regulation of yabG transcription could be simplified considerably.

Western blots in Figures S2 and S8 are over-exposed/ over-contrasted so that it looks like the background has been washed out; western blot in Fig S13 is a more appropriate exposure.

The statement on line 271 that “release of the CspB domain from CspA is necessary for the activation” may be true, but I don't think it has been formally shown because loss of YabG or mutation of the main YabG cleavage site does not prevent interdomain processing of CspBA – the cleavage site just moves to a different location

**Part II – Major Issues: Key Experiments Required for Acceptance**

Reviewer #2: (No Response)

**Part III – Minor Issues: Editorial and Data Presentation Modifications**

Reviewer #2: Minor comments:

Line 44: Remove the “m” from Clostridioidesm”

Line 54: consider adding “which is” before “required”

Line 69: consider deleting “as during antibiotic treatment”

Line 115: change “spore” to “spore’s”

Line 339: what does “congenic” mean? – do the authors mean “isogenic”?

Line 354: consider using “similar” instead of “not significantly different”

Line 532: would be better for yabG to refer to the protein rather than the gene here

Line 538: please delete “is”

Line 773: consider changing “promoters” to “transcriptional regulation

PLOS authors have the option to publish the peer review history of their article (what does this mean?). If published, this will include your full peer review and any attached files.

Reviewer #2: No

---

## [Editor Report · Acceptance letter]

7 Nov 2023

Dear Prof Henriques,

We are delighted to inform you that your manuscript, "A sporulation signature protease is required for assembly of the spore surface layers, germination and host colonization in <i>Clostridioides difficile<i>," has been formally accepted for publication in PLOS Pathogens.

Best regards,

Kasturi Haldar

Editor-in-Chief

PLOS Pathogens

orcid.org/0000-0001-5065-158X

Michael Malim

Editor-in-Chief

PLOS Pathogens

orcid.org/0000-0002-7699-2064